# Broadly neutralizing antibodies isolated from HEV convalescents confer protective effects in human liver-chimeric mice

George Ssebyatika [1,14], Katja Dinkelborg [2,3,4,14], Luisa J. Ströh[5], Florian Hinte[4,6], Laura Corneillie[7], Lucas Hueffner [2], Elina M. Guzman [1], Prossie L. Nankya[1], Nina Plückebaum [5], Lukas Fehlau [2], Jonathan Garn[2], Nele Meyer[2], Sarah Prallet[8], Ann-Kathrin Mehnert [8], Anke R. M. Kraft [3,4,9], Lieven Verhoye[7], Carina Jacobsen [5], Eike Steinmann[10], Heiner Wedemeyer[3,4,11], Abel Viejo-Borbolla [5,11], Viet Loan Dao Thi [4,8], Thomas Pietschmann [2,4,11], Marc Lütgehetmann [4,12], Philip Meuleman[7], Maura Dandri [4,6], Thomas Krey [1,4,5,11,13] ✉ & Patrick Behrendt [2,3,4] ✉

Hepatitis E virus (HEV) causes 3.3 million symptomatic cases and 44,000 deaths per year. Chronic infections can arise in immunocompromised individuals, and pregnant women may suffer from fulminant disease as a consequence of HEV infection. Despite these important implications for public health, no specific antiviral treatment has been approved to date. Here, we report combined functional, biochemical, and X-ray crystallographic studies that characterize the human antibody response in convalescent HEV patients. We identified a class of potent and broadly neutralizing human antibodies (bnAbs), targeting a quaternary epitope located at the tip of the HEV capsid protein pORF2 that contains an N-glycosylation motif and is conserved across members of the *Hepeviridae*. These glycan-sensitive bnAbs specifically recognize the non-glycosylated pORF2 present in infectious particles but not the secreted glycosylated form acting as antibody decoy. Our most potent bnAb protects human liver-chimeric mice from intraperitoneal HEV challenge and co-housing exposure. These results provide insights into the bnAb response to this important emerging pathogen and support the development of glycan-sensitive antibodies to combat HEV infection.

The hepatitis E virus (HEV), a quasi-enveloped single-stranded RNA virus, belongs to the genus *Paslahepevirus* within the family *Hepeviridae* and is the most common cause of viral hepatitis worldwide with an estimated 20 million annual infection events, at least 3.3 million symptomatic cases, and 44,000 deaths[1]. Overall, four major genotypes are pathogenic for humans (GT 1–4). Of these, GT 1 and 2 are solely circulating in humans, while GT 3 and 4 are zoonotic viruses that are responsible for relevant sporadic and autochthonous infections in humans caused mainly by consumption of undercooked pork[2,3]. HEV

usually causes a self-limiting disease, however, GT 1 infections can cause up to 30% mortality in pregnant women in the third trimester[4], and chronic GT 3 infections can occur in immunocompromised individuals, such as those receiving organ transplants or chemotherapy and individuals with HIV infection[5–7]. Chronic infection can lead to the rapid progression of liver fibrosis and cirrhosis with the ultimate need for liver transplantation[8]. Of note, rat HEV, belonging to the genus *Rocahepevirus* and displaying only ~55–60% amino acid (aa) identity to GTs 1–4, has recently been reported to cause chronic as well as acute

infections in a substantial number of individuals[9–11]. To date, no specific antiviral therapy exists. In chronic infections, an off-label therapy with ribavirin for several months shows response rates of up to 85%[12], but severe side effects such as hemolytic anemia render this option difficult to manage, and not all patients are eligible (e.g., due to severe renal insufficiency). Therefore, novel antiviral strategies targeting both acute and chronic HEV infections are urgently required. An efficient HEV vaccine based on recombinant virus-like particles consisting of a truncated version of the capsid protein pORF2[13] is available and induces protective humoral immune responses preventing symptomatic GT 4 infections but is only licensed in China and Pakistan.

The positive-strand RNA HEV genome comprises 7.2 kb and encodes three open reading frames (ORFs), of which *ORF2* encodes the viral capsid protein containing a signal peptide at its N-terminus that leads to translocation into the secretory pathway. *ORF2* translation via an alternative start codon that omits the N-terminal signal peptide gives rise to cytosolic pORF2 that assembles into viral capsids[14]. HEV is released from infected cells with an outer lipid layer formed by intracellular membranes[15], which is not equivalent to the envelope of canonical enveloped viruses such as flaviviruses or Influenza viruses. Bile acids are believed to dissolve this bilayer membrane upon release into the biliary tract, giving rise to naked virions in stool that play an important role in spread of the virus[16]. The "quasi-envelope" surrounds a capsid following a pseudo T = 3 icosahedral symmetry with a diameter of approximately 30–33 nm[17–19], which consists of the capsid protein (pORF2) and identifies the pORF2 protruding (P) domain as the dimeric spike on the surface of the capsid. Atomic resolution models exist only for recombinant virus-like particles (VLPs) but not for bona fide infectious viruses.

Antibodies represent a versatile option to combat infectious diseases, cancer, as well as autoimmune and chronic inflammatory diseases[20]. Approved antibody therapies are available for several virus infections including the respiratory syncytial virus[21–23], Ebola virus[24] and more recently severe acute respiratory syndrome coronavirus 2 (SARS-CoV-2; reviewed in ref. [25]). First insights into antibody-mediated HEV neutralization were gained from the identification of murine neutralizing antibodies (nAbs) targeting the pORF2 P domain, and crystal structures of the respective antibody fragments in complex with a soluble P domain dimer[26,27]. NAb 8C11 binds to a conformational epitope at the side of the P domain dimer perpendicular to the dimeric interface in a genotype-specific manner[27]. In contrast, the broadly neutralizing antibody (bnAb) 8G12 binds to a non-overlapping quaternary epitope at the side of the P domain dimer directly at its dimeric interface[26]. In this context, it is important to note that available structural data on HEV VLPs highlight an exposed N-linked glycosylation site within pORF2 termed N3 ($N_{562}$;[28]), which is positioned at the dimeric interface of the pORF2 P domain apex.

HEV has developed two efficient strategies to evade the humoral immune response. The originally suggested *ORF2* start codon is in frame with a 15 amino acids signal peptide, giving rise to the secretion of a presumably dimeric pORF2 form that is glycosylated during the passage through the secretory pathway and has been detected in high concentrations (~200 ng/ml) in serum of infected macaques[14]. These pORF2 dimers are believed to act as decoy for neutralizing antibodies and differ in their glycosylation status from pORF2 present in infectious particles (reviewed in ref. [29]). As a second strategy, the complex assembly pathway of HEV particles located in the cytoplasm results in the formation of the "quasi-envelope" described above, which may hinder direct antibody binding to the viral capsid and therefore likely limits neutralization.

Based on available structural insights as well as on combined results from antibody competition and alanine scanning using a panel of representative murine monoclonal antibodies (mAbs), a detailed epitope mapping of pORF2-specific antibodies revealed six potential conformational and functional antigenic sites on the P domain dimer[30].

Following this approach, a panel of vaccine-induced human mAbs isolated from four vaccinated patients was recently reported to interact with five distinct antigenic sites on the P domain[31], with the most efficient nAbs binding to the side of the P domain dimer.

Here, we describe a class of potent glycan-sensitive bnAbs from HEV convalescent patients. These antibodies target pORF2, the viral receptor-binding protein and a major target of neutralizing antibodies. We report a detailed characterization of their breadth and neutralization capacity, revealing that binding to the glycosylated pORF2 dimer is hampered by the presence of a glycan at $N_{562}$. X-ray crystallographic analyses showed that the epitopes targeted by these antibodies involved $N_{562}$, located within a highly conserved region at the pORF2 P domain dimer apex. Neutralization by these glycan-sensitive antibodies is unaffected by the soluble pORF2 dimer that likely acts as decoy in HEV-infected patients, whereas the neutralizing capacity of potent glycan-insensitive bnAbs is markedly reduced by this antibody decoy. In vivo analysis in human liver-chimeric mice revealed a strong protective activity of our most potent glycan-sensitive antibody from HEV challenge. Finally, we provide evidence on the frequency of such glycan-sensitive antibodies in acutely HEV-infected patients. In summary, our results provide important insights to understand the human antibody response to this emerging human pathogen, shed light on a broadly conserved epitope that will inform vaccine design and highlight glycan-sensitive antibodies as promising protective and therapeutic tools for therapy of HEV infection.

## Results

### Generation of human nAbs targeting HEV pORF2 from convalescent patients

Using Magnetic Activated Cell Sorting (MACS) and flow cytometry we isolated single HEV-reactive memory B cells from two individuals, p60 and p61, who cleared an acute, clinically apparent HEV infection approximately 12 months earlier. 3.5 and 4.1 % of CD19 + /CD20 + /IgG+ B cells per individual, respectively, specifically interacted with a non-glycosylated green fluorescently labeled HEV GT 3 pORF2 P domain dimer (amino acids 456–660, Fig. S1), and 2.8 and 2.1% of CD19 +/ CD20 + /IgG+ B cells, respectively, specifically interacted with a red fluorescently labeled HEV pORF2 P domain dimer (Fig. S2) identical in sequence, but secreted into the supernatant due to the presence of an N-terminal signal peptide and therefore carrying a glycan at position N562 (Fig. S1). These two populations were combined with B cells reacting with both P domain forms (8.1 and 8.4 % of CD19 +/CD20 + / IgG+ B cells per patient, respectively, Fig. S2), yielding a total of 23,934 P domain-specific memory B cells, of which 3,364 productively paired heavy and light-chain sequences passed the quality control (342 and 3022 per individual, respectively). Sequence analysis revealed a polyclonal response, mainly of the IgG1 and IgG2 subclasses (Fig. 1a) with few clonally related sequences per individual (~2–7%) and no clonotypes identified more than thrice (Fig. 1b). Notably, antibodies from donor p61 predominantly used the kappa light chain, whereas we observed no obvious preference for donor p60 (Fig. 1c).

The recombinant HEV vaccine (Hecolin®) that protected against symptomatic HEV GT 4 infection in a large phase III clinical trial in China[13] induced pORF2-specific mAbs of different $V_H$ germlines with IGHV1-69 predominantly activated, and to a lower extent IGHV1-18, IGHV3-30, and IGHV4-59[31]. We observed patient-specific differences with no obvious dominating germline genes (Fig. 1d, e), but IGHV3-30, IGHV3-30-3, IGHV3-23, and IGHV1-69 were the most activated germline genes for donor p61, whereas IGHV6-1, IGHV5-51, and IGHV4-59 frequently appeared for donor p60 (Fig. 1d). The CDRH3 length in antibodies from both individuals ranged from 5 to 26 amino acids with an average length of about 15 amino acids (Fig. 1f, Table S1), although minor differences between individual patients were observed, (e.g., a CDRH3 length ≤13 amino acids was more abundant in p60) and the identity of the VH genes and Vκ/Vλ genes with a putative germline

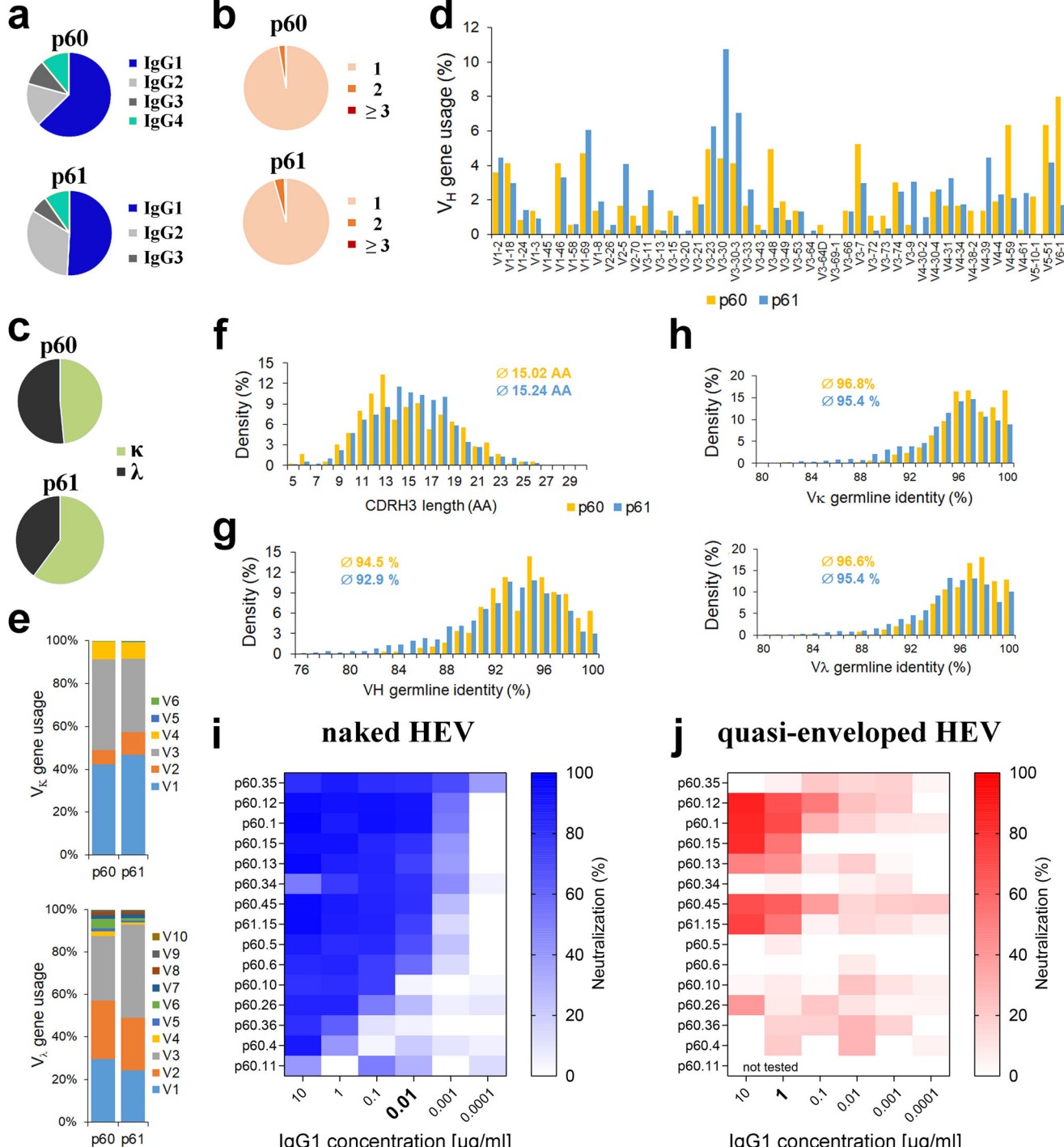

**Fig. 1 | B cells of convalescent HEV patients encode potent nAbs.** HEV-specific B cell repertoire analysis from two donors showing the distribution of productive IgG subclasses (**a**), clonality within individual patients (**b**), and type of light chains (**c**) as well as heavy (**d**) and light chain (**e**) variable (V) gene usage. Frequencies of heavy chain complementarity determining region 3 (CDRH3) length (**f**) and % germline identity of heavy chain variable segment (VH) (**g**) and light chain variable segment (VH)

Vκ/Vλ (**h**) are shown. Mean values for CDRH3 length and mean germline identity are given. (i + j) The heat maps show neutralization in % as measured by focus forming units per well against the naked (**i**) and quasi-enveloped (**j**) particles of the HEV GT 3 Kernow-C1 p6 G1634R strain. (*n* = 3 biological replicates). Source data are provided within the Source Data file.

precursor gene ranged from 80–100% (Fig. 1g, h). Other B-cell repertoire analyses have reported similar average lengths for the CDRH3[32,33].

To functionally characterize a panel of monoclonal human antibodies from HEV-infected patients, we cloned and expressed 85 heavy- and light-chain paired sequences in the form of single chain variable fragments (scFv) in *Drosophila melanogaster* S2 cells[34]. A single-dose neutralization assay using these scFvs together with naked HEV virions

based on a recently established HEV GT 3 cell culture system[35] was used to identify the most potent neutralizing scFv (Fig. S3). In line with the germline usage analysis of P domain-specific B cells, germline usage analysis of neutralizing scFv revealed no evident bias towards particular germline genes, e.g., as described for the germline VH1-69 in patients infected with the hepatitis C virus[33]. The four most potent scFv in neutralization of both naked and quasi-enveloped virions (p60.1,

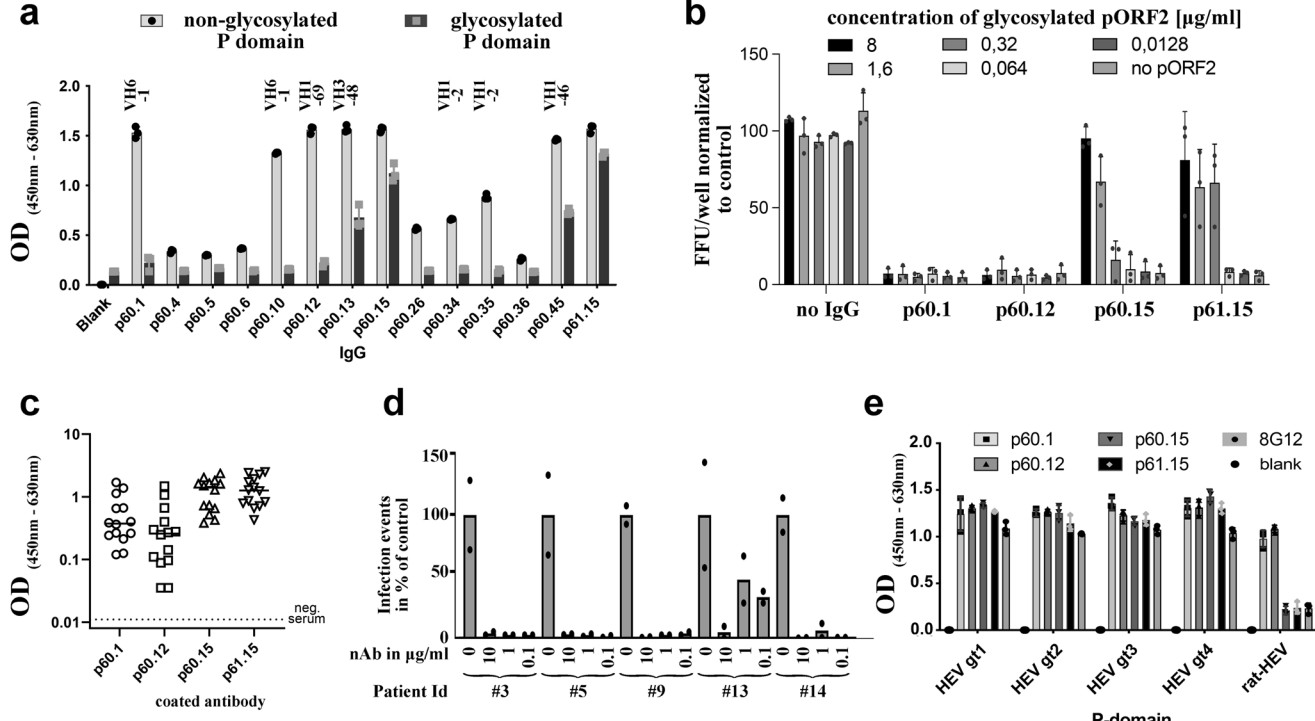

**Fig. 2 | Characterization of pORF2 nAbs. a** IgG antibody binding to a non-glycosylated HEV GT 3 P domain purified from the cytoplasm (light gray) and an identical P domain glycosylated in the secretory pathway (dark gray; *n* = 3 biological replicates). Variable heavy chain germline segments (VH) of glycan-sensitive antibodies are indicated above the bar plot. **b** Neutralization of naked viral particles of HEV Kernow-C1 p6 G1634R using 0.1 μg/ml of the respective nAb after pre-incubation with increasing concentrations of glycosylated P domain. The graph depicts the number of infected cells normalised to the control. (*n* = 3 biological replicates) **c** ELISA analysis on plasma of 14 viraemic patients reveals that all tested nAbs bind to patient isolate antigens. (*n* = 3 biological replicates). Overall, glycan sensitive nAbs show approximately 10x lower optical density (OD) values. **d** Clinical

HEV isolates are efficiently neutralized by bnAb p60.1. A total of five stool filtrates from HEV patients were used to infect stem-cell derived hepatocyte-like cells with or without p60.1 at indicated concentrations (*n* = 2 biological replicates). 7 days post-infection, HEV infection was analyzed by quantifying pORF2-positive cells using Zen software. **e** IgG antibody binding to a non-glycosylated HEV P domain derived from the indicated HEV genotype demonstrates broad reactivity for all four tested nAbs (*n* = 3 biological replicates). Only glycan-sensitive nAbs also recognize the P domain derived from rat HEV. All ELISA experiments shown in panel a, b and e were performed at least in technical triplicates. Source data are provided within the Source Data file. Error bars represent the standard deviation.

p60.12, p60.15, p61.15) belonged to four different germline genes and contained CDRH3 sequences of 11–17 amino acids, similar to the overall pORF2-specific CDRH3 length distribution in both individuals (Fig. 1f).

The 15 most potent scFv were then expressed as intact IgG1 isotypes in HEK Expi293F cells, and their neutralization capability was evaluated. Ten of the fifteen mAbs neutralized the naked version of the HEV GT3 Kernow-C1 p6 G1634R strain by more than 60% at a concentration of 10 ng/ml (Fig. 1i, highlighted in bold). Additionally, several mAbs also neutralized quasi-enveloped virions of the same viral strain, although approximately 100-fold higher concentrations were needed to reach similar neutralization efficiencies, with five antibodies showing neutralization of more than 50% at a concentration of 1 μg/ml (Fig. 1j, highlighted in bold).

**A subset of HEV nAbs interacts with pORF2 in a glycan-sensitive manner**

We used recombinant HEV P domain expressed in the cytosol (non-glycosylated at $N_{562}$) along with its secreted counterpart (glycosylated at $N_{562}$; Fig. S1) to assess the reactivity of nAbs to both proteins in ELISA. We observed a large range of signal strengths that possibly reflects variation in binding affinities to the recombinant P domain. However, several nAbs recognized both P domains to a similar extent (e.g., p60.15 or p61.15; Fig. 2a), whereas other nAbs, such as p60.1 and p60.12, efficiently recognized the non-glycosylated P domain, but did not interact with the glycosylated one (Fig. 2a), suggesting that the

presence of the glycan sterically interfered with antibody binding, i.e., a glycan-sensitive binding, and thus likely the glycosylation motif is either juxtaposed or directly part of the epitope.

A total of five different heavy chain germline genes gave rise to the identified glycan-sensitive antibodies (Fig. 2a), suggesting that this group of antibodies do not share a common genetic signature. Glycan-sensitive antibodies instead develop from different germline precursors and were induced in both tested individuals. Previously, human mAbs directed against the previously defined site C3 surrounding the N-linked glycosylation site at the apex of the P domain dimer[30] were found only in two out of four donors and exhibited weak or no neutralization capacity[31], suggesting that such antibodies are not observed frequently and do not play an important role for virus neutralization. In our study, we also identified weaker nAbs likely targeting the glycosylation site $N_{562}$ such as p60.10, in line with this recent observation[31]. Further research is required to elucidate functional differences in comparison to the highly potent glycan-sensitive nAbs isolated in this study.

The four nAbs most potently neutralizing quasi-enveloped and naked virions (two glycan-sensitive (p60.1 and p60.12), and two glycan-insensitive nAbs (p60.15 and p61.15)) bound the non-glycosylated P domain of GT3 HEV with picomolar affinity, as measured by surface plasmon resonance (SPR; Fig. S4, Table S2). The affinity of the glycan-insensitive nAbs p60.15 and p61.15 to the glycosylated P domain was 2-6-fold lower, likely because the attached glycan interfered with epitope accessibility. In line with the ELISA

results described above, no kinetic binding parameters could be determined for nAbs p60.1 and p60.12 to the glycosylated P domain (Table S2).

The glycosylated pORF2 dimer in patient sera has been proposed to serve as antibody decoy[14], but this decoy might not affect the neutralization potency of glycan-sensitive nAbs like p60.1 or p60.12, as they do not bind the glycosylated P domain.

To further address this decoy hypothesis, we preincubated glycan-sensitive (p60.1 and p60.12) and glycan-insensitive (p60.15 and p61.15) antibodies with recombinant dimeric glycosylated pORF2 (amino acids 112-660) expressed in HEK Expi293F cells. Subsequent neutralization assays with preincubated antibodies revealed that the glycosylated pORF2 dimers block neutralization of preincubated glycan-insensitive antibodies, but not glycan-sensitive antibodies in a dose-dependent manner (Fig. 2b). This specificity of glycan-sensitive nAbs for non-glycosylated pORF2 renders this antibody class prominent candidates for clinical development. This also implies that both our glycan-sensitive and glycan-insensitive antibodies constitute unique tools to determine levels of circulating glycosylated dimer[14] and infectious particle-associated pORF2 to understand their physiological role in acute and chronic HEV infection.

## HEV nAbs neutralize primary isolates and are cross-reactive to different genotypes

A neutralization assay using a second HEV GT 3 strain (83-2, amino acid identity to the p6 strain in the P domain of 96.1%) revealed a less pronounced neutralization across most antibodies than observed for the p6 strain, suggesting differences in the assay efficiency. This neutralization assay provides first evidence for a broader, isolate-independent neutralizing activity of several antibodies (Fig. S5). This is an important consideration because of the potential occurrence of escape mutations that could impair the efficiency of antibodies against viral variants. Such an immune escape has regularly been observed for SARS-CoV-2 to escape the immune pressure exerted by the antibody response[36,37]. To determine whether broad reactivity also extends to patient-derived isolates we quantified binding of individual antibodies to antigen from serum of 14 chronically infected HEV RNA-positive patients who likely had an HEV GT 3 infection as this is the most abundant GT in the study area (none of the participant had a history of traveling to endemic regions). As expected[38], these sera contained high amounts of pORF2 (Table S3, patients 1–14) and all four potent nAbs recognized pORF2 in sera from most patients. As all patients were viremic, two forms of pORF2 were expected to be present, including (1) non-glycosylated pORF2 in form of infectious capsids and (2) glycosylated pORF2 dimers likely serving as antibody decoy, with the latter representing the much more abundant form[38]. This was indeed reflected in the on average ~10-fold higher OD-values obtained with glycan-insensitive nAbs when compared with glycan-sensitive nAbs (Fig. 2c).

In certain patients infected with HEV, pORF2 can be detected after viral RNA has been cleared from both serum and stool during antiviral treatment for chronic infection[39]. In our study, we identified two such patients by analyzing longitudinal samples collected before, during, and after treatment with ribavirin. Interestingly, our ELISA using the glycan-sensitive antibody bnAb p60.1 showed kinetics that aligned well with detection of viral RNA, the current gold standard for detecting viremia (Table S7). In contrast, an analogous ELISA using the glycan-insensitive bnAb p60.15 as well as the widely used HEV Ag WANTAI assay remained positive even in samples that were already negative for HEV RNA (Table S7). Given the similar avidity determined for bnAbs p60.1 and p60.15 (Table S2), this observation likely indicates the detection of residual circulating glycosylated pORF2 dimers after viral clearance.

To analyze the impact of non-adapted HEV isolates on infection, we measured the neutralization potential of our most potent nAb p60.1 on human induced pluripotent stem cell (iPSC)-derived hepatocyte-like cells (HLCs;[40]). These cells, unlike cancerous hepatoma cells, are readily permissive for non-adapted and authentic HEV infection[40]. A neutralization assay using primary HEV isolates from patient stool samples on HLCs revealed a potent neutralization of viruses derived from all five patients, even at the lowest concentration applied (Fig. 2d). Our most potent nAbs also demonstrated a high signal-to-noise ratio in an indirect immunofluorescence assay in HEV-GT 3-transfected HepG2 cells (Fig. S6).

We extended our analysis of the breadth of our most potent nAbs (p60.1, p60.12, p60.15 and p61.15) to a larger variety of members of the family *Hepeviridae* using a panel of five recombinant P domains derived from GT 1 to 4 isolates belonging to the genus *Paslahepevirus* and a rat HEV isolate belonging to the genus *Rocahepevirus*. All four nAbs bound the P domains derived from the human pathogenic genotypes (GT 1–4, Fig. 2e) in ELISA, similar to the murine bnAb 8G12 used as control[26]. The glycan-insensitive nAbs p60.15 and p61.15 as well as the control bnAb 8G12 did not interact with the P domain of the rat HEV isolate. In contrast, both glycan-sensitive nAbs also recognized the evolutionary more distant rat HEV P domain, in line with previous reports demonstrating cross-reactive antibodies targeting the capsid protein[41]. Kinetic binding parameters of nAb-binding to these different pORF2 domains, measured using SPR, revealed a picomolar binding of all four nAbs to P domains of all four human pathogenic GTs. The equilibrium dissociation constant of p60.1 and p60.12 to the rat HEV P domain was in the micromolar range (Table S4), exhibiting an approximately 1000-fold lower affinity than the other four P domains. These results substantiate that the epitope targeted by the glycan-sensitive bnAbs is highly conserved among members of *Hepeviridae*, implying that viruses carrying specific escape mutations are not expected to readily emerge.

## Crystal structures of HEV bnAbs in complex with a P domain dimer

A more detailed view of the targeted epitopes was obtained from crystal structures of p60.1, p60.12, p60.15 and p61.15 antibody fragments in complex with a GT 3 P domain ranging in resolution from 1.91 Å to 2.4 Å and a rat HEV P domain in complex with a p60.12 Fab fragment at a resolution of 3.8 Å (Fig. 3, 4, Fig. S7, S8, Supplementary Table S5).

Both bnAbs p60.15 and p61.15 recognize partially overlapping epitopes located in and around a groove zone on a single pORF2 protomer (Fig. 3a, b). The interactions of the P domain with p60.15 and p61.15 bury a total of 785.2 and 857.2 Å$^2$ of the surface area, respectively, as calculated by PISA, with approximately 60.5 and 54.9% (or 514.1 and 471 Å$^2$) buried by the heavy chain, respectively. The epitopes partially overlap with the previously described antigenic sites C5 and C6[31] and share contact residues E479, D496, R512, and R578. Comparison with the epitope targeted by the murine nAb 8C11[27] reveals a strong overlap with bnAb p61.15 (Fig. 3d–f), although the two nAbs differ in breadth; p61.15 recognizes all four GTs while 8C11 interacts with GT 1 in a GT-specific manner. BnAb p60.15 targets an epitope located at the top of the groove zone, which contains residues that play an important role for interaction with neutralizing antibodies[30].

In contrast, p60.1 and p60.12 recognized a quaternary epitope that assembles mainly from conserved residues in the dimerization interface at the tip of the P domain dimer (Fig. 4, Figs. S7, S11). Both bnAbs bind predominantly to one protomer but also make contacts with the other one, resulting in an asymmetric binding mode and indicating that a correctly folded pORF2 dimer is required for antigen-antibody interaction. The interactions of the P domain with p60.1 and p60.12 bury a total of 785.9 and 885.6 Å$^2$ of the surface area, respectively, as calculated by PISA, with approximately 71.3 and 58.6% (or 560.3 and 519 Å$^2$) buried by the heavy chain, respectively. Both epitopes are located within antigenic site C3 and largely overlap, sharing

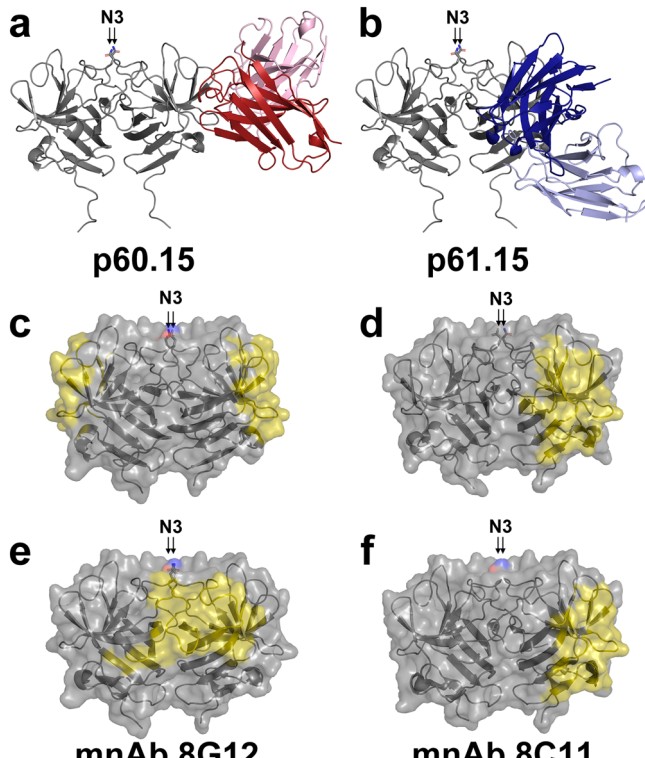

**Fig. 3 | Crystal structure of P domain in complex with glycan-insensitive bnAbs.** Cartoon view of complex structures of the GT3 P domain (gray) with p60.15 (red; **a**) and p61.15 scFv (blue; **b**), which recognize partially overlapping epitopes on the P domain dimer (gray); only the bnAb fragment binding to the right P domain pro-tomer is shown for clarity. The position of the N-linked glycosylation site N3 is indicated. **c–f** Side-view of the footprints (yellow) of human "glycan-insensitive" bnAbs on the P domain dimer (gray) in comparison to described murine nAbs 8G12 (PDB 4PLJ;[26]; **e**) and 8C11 (PDB 3RKD;[27]; **f**). 8G12 is the only antibody that binds at the P domain dimer interface. Sidechain oxygen and nitrogen atoms of $N_{562}$ are colored in red and blue, respectively, to indicate the position of N3.

contact residues $Q_{482}$, $R_{488}$, $Y_{557}$, $Y_{561}$, $N_{562}$ and $Y_{584}$. Importantly, the $N_{562}$ side chains of both protomers, which serve as attachment points for the N-linked glycans in the secreted pORF2 dimer, are buried in the interface with both antibodies. Due to this direct involvement of $N_{562}$ into the interaction, an attached glycan must sterically block binding of these bnAbs, explaining the observed glycan sensitivity. Mapping of amino acid conservation of the P domains from the human pathogenic HEV GTs and the distantly related rat HEV reveals that the epitopes recognized by the two potent glycan-sensitive bnAbs p60.1 and p60.12 are located at the tip of the P domain in an area that is highly conserved across *Hepeviruses* (Figs. S7B, S11). Indeed, a crystal structure of the p60.12 Fab fragment in complex with a rat HEV P domain confirmed its binding mode to a quaternary epitope at the tip of the P domain dimer (Fig. S7C).

The observed binding mode also explains the specific recognition of the infectious HEV particle that is not affected by the secreted pORF2 dimer decoy (Fig. 2b). To estimate the prevalence of glycan-sensitive antibodies during HEV infection, we analyzed sera from three patients that cleared an acute, clinically apparent HEV infection within 12 months. Comparative depletion analysis using either glycosylated or non-glycosylated P domain revealed that all three patient sera contained considerable amounts of IgG targeting the P domain. In all three individuals approximately 20% of these anti-P domain antibodies were not depleted by glycosylated P domain, suggesting that an acute HEV infection induces considerable amounts of glycan-sensitive antibodies in immunocompetent patients (Fig. S9). Further analyses will be

required to decipher the role of glycan-sensitive antibodies in acute and chronic HEV infection.

## BnAb p60.1 protects humanized mice from HEV infection

Several studies demonstrated that human liver chimeric mice can be infected with HEV and are useful tools for studying HEV infection[35,38,42–45]. The protective potential of the most potent glycan-sensitive human HEV nAb was determined in vivo by pre-treating six human liver chimeric mice with antibody p60.1 or a PBS control 24 h prior to an intraperitoneal challenge with cell culture-produced, naked GT 3 HEV and a second treatment with antibody or PBS on day 4.

As depicted in Fig. 5a, high copy numbers of HEV RNA were detectable in the feces of all infected control animals over several weeks without decline. In contrast, in feces of antibody-treated mice HEV RNA was not detected throughout the first four weeks, demon-strating a protective effect against HEV challenge. In only one out of three antibody-treated mice, low levels of HEV RNA became detectable from week four on, albeit remaining ~1000x lower than in control mice. NGS of the virus isolated from this mouse revealed an unaltered sequence encoding the p60.1 epitope. Such a breakthrough event has been described for passive immunization experiments against the hepatitis B and C viruses[46,47], demonstrating successful initial infection and the high level of viral challenge these immunodeficient animals were subjected to. The delay in virus shedding also indicates that – in contrast to the control animals - the spread of the virus was efficiently controlled as long as p60.1 was circulating in serum (Fig. 5a). Virus shedding (judged from excreted viral RNA in stool) was detected only after the decline of human antibody in serum.

In parallel, we assessed the capacity of anti-HEV nAb p60.1 to block de novo GT 1 infection in a more complex, clinically relevant setting, where the infection challenge occurs through the fecal-oral route. In this co-housing setting, four mice received p60.1 and five received Palivizumab, a control antibody, while five additional mice, stably infected with a patient-derived HEV GT 1 strain, served as infection source. Mice received the respective antibodies (1 mg) 24 h before starting the co-housing with HEV RNA-positive animals and 0.5 mg antibody was injected on days 3 and 7. After 12 days of co-housing, mice were separated and HEV development was monitored for additional 7 weeks. The results showed that mice treated with p60.1 were only transiently positive in feces during co-housing (demon-strating contact and uptake of stool-derived HEV; Fig. 5b), but feces of all 4 animals became promptly negative in the week after separation. In contrast, control-treated mice first showed a decrease of HEV loads in feces (due to the separation from high-titer HEV-positive mice) and then HEV levels increased in feces and mice became positive also in serum (Fig. S10A), demonstrating establishment of infection. Immu-nofluorescence staining on liver sections from donor mice (sacrificed on day 13) and control antibody-treated mice (sacrificed on day 62) revealed pORF3 staining, mainly located at the apical membrane of the human hepatocytes. In contrast, no pORF3 signal was detected in liver sections of a p60.1-treated mouse (sacrificed on day 62; Fig. S10B), in line with the lack of detectable viral genome in feces and serum (Fig. 5 and Fig. S10A).

An important limitation of these in vivo studies is that the route of infection and spreading of HEV into human livers are poorly under-stood to date, as it remains elusive whether naked ingested viruses target a primary site of replication within the gastrointestinal tract (reviewed in ref. 48) or rather "transit" the gut to reach the liver. The situation in human liver-chimeric mice, which lack human enterocytes susceptible to HEV infection but can be infected through the fecal-oral route, is likely even more complex. It is therefore difficult to assess, whether nAb p60.1 protected the mice from infection via two distinct infection routes using a single shared mechanism or two distinct mechanisms. Future experiments will aim to elucidate the exact route and critical determinants for establishing an infection in these mice

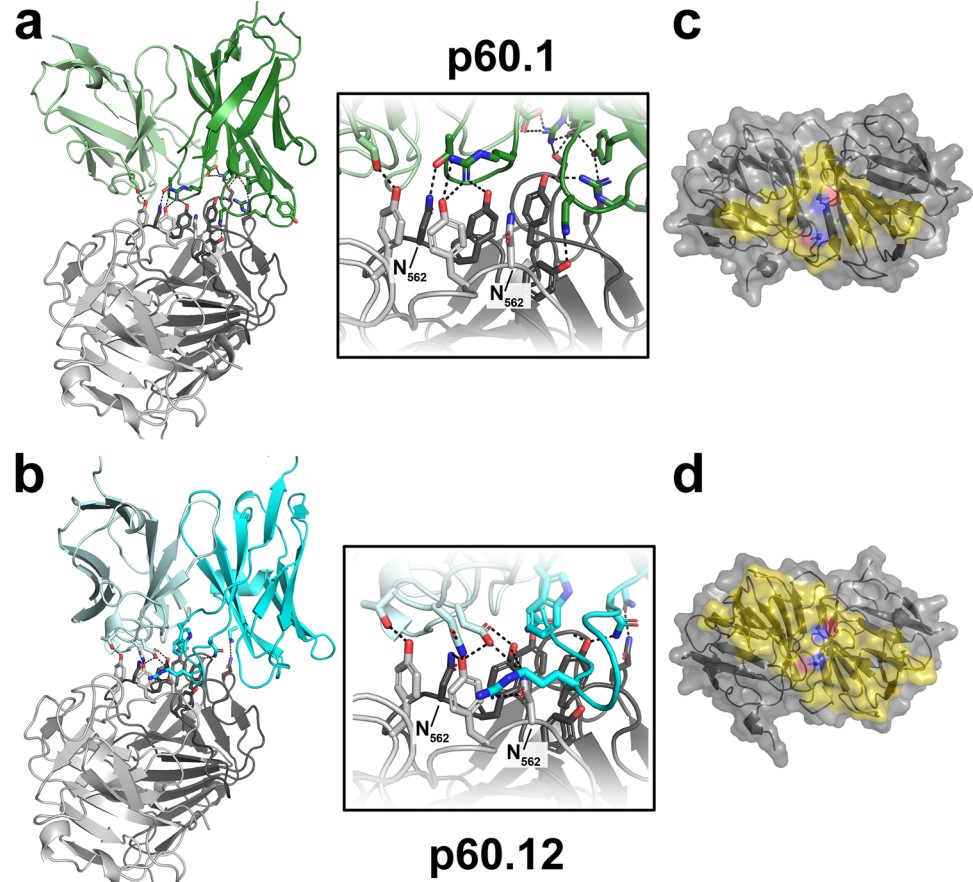

**Fig. 4 | Crystal structure of P domain in complex with glycan-sensitive bnAbs.** Cartoon view of P domain structures in complex with p60.1 (green; **a**) and p60.12 (cyan; **b**) that recognize partially overlapping epitopes at the tip of the P domain dimer (individual protomers colored in light and dark gray). Same view as in Fig. 3a, b. The insets show zoomed views on the interface with a particular focus on the two sidechains of $N_{562}$. Sidechain oxygen and nitrogen atoms are colored in red and blue, respectively, dashed lines indicate the complex hydrogen binding network stabilizing the high affinity interaction. **c** + **d** Top-view of the footprints of human glycan-sensitive bnAbs on the P domain dimer (gray). Sidechain oxygen and nitrogen atoms of $N_{562}$ are colored in red and blue, respectively, to indicate the position of N3.

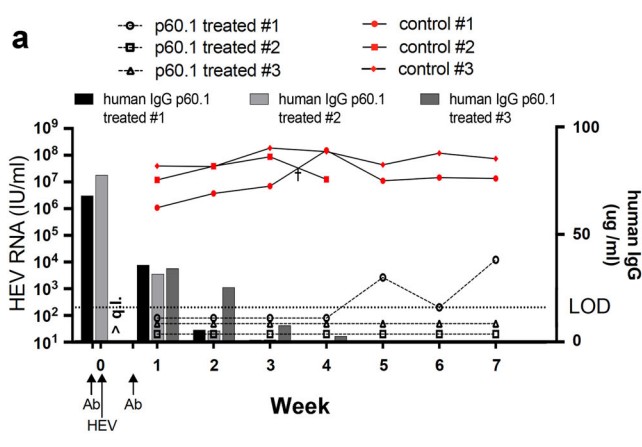

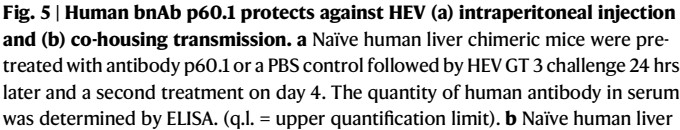

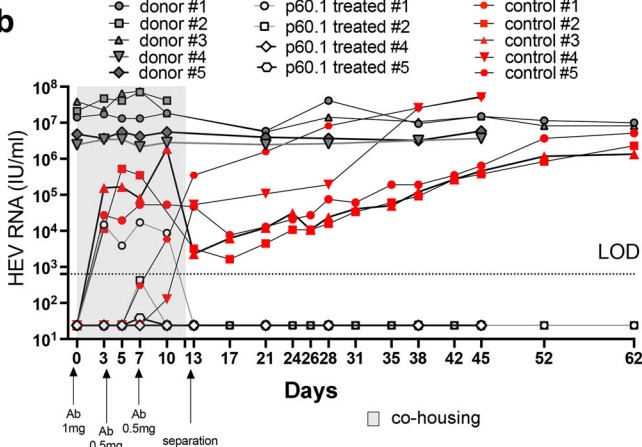

**Fig. 5 | Human bnAb p60.1 protects against HEV (a) intraperitoneal injection and (b) co-housing transmission. a** Naïve human liver chimeric mice were pre-treated with antibody p60.1 or a PBS control followed by HEV GT 3 challenge 24 hrs later and a second treatment on day 4. The quantity of human antibody in serum was determined by ELISA. (q.l. = upper quantification limit). **b** Naïve human liver chimeric mice were pre-treated with antibody p60.1 or an irrelevant control antibody prior to 12 days of co-housing with HEV GT 1-infected human liver chimeric mice. In both cases, HEV infection of individual mice was monitored by RT-PCR to detect HEV genomes in feces. Source data are provided within the Source Data file. LOD lower limit of detection.

and their relevance to the situation in humans. Only then, a precise comparative assessment of the protective effects of the glycan-sensitive and glycan-insensitive antibodies in vivo can be conducted.

Although major differences between the route-of-infection used in the two animal experiments may exist, all animals were initially infected with naked HEV particles, similar to the primary infection in exposed humans. The glycan-sensitive antibody p60.1 has a strong protective effect against HEV challenge in both cases, independent of the infecting HEV GT, rendering this antibody an excellent candidate for clinical applications.

## Discussion

In this study, we combine antigen-specific single-cell sorting, next-generation sequencing, structural biology, and functional and bio-chemical antibody characterization in vitro and in vivo to understand the antibody response induced by acute HEV infection in humans. Our results provide important insights for the informed design of a cross-protective HEV vaccine and highlight the potential of glycan-sensitive human antibodies as promising tool to prevent and possibly treat HEV infection. The discovery of such specific antiviral agents is an important step in development of antiviral treatments for pregnant patients and chronically as well as acutely infected individuals who do not respond to or are not eligible for ribavirin treatment.

We observed a trend that most nAbs that potently neutralized naked particles also neutralized purified quasi-enveloped particles, in contrast to previous reports[49]. We can only speculate that antibody characteristics like epitope specificity and kinetic binding parameters of those antibodies might be important for efficient neutralization of quasi-enveloped particles. Such an efficient neutralization of quasi-enveloped virions is in line with studies on Hepatitis A virus (HAV), which also circulates in the bloodstream as quasi-enveloped virions. HAV is effectively neutralized by antibodies acting during the late stages of viral entry, specifically in the late endosomal compartment, where the capsid is stripped from its membrane and becomes accessible to neutralizing antibodies[48]. A similar post-attachment neutralization of HEV would also be consistent with the stable protection provided by the humoral immune response in the in vivo scenario, as observed in currently established vaccine strategies[50]. Further studies will be required to address the mechanism how quasi-enveloped particles are efficiently neutralized by bnAbs.

Using monoclonal antibodies as a therapeutic option against HEV presents a promising approach, since the virus - despite its relatively high mutation rate - predominantly exists as a single serotype, enhancing its susceptibility to antibody-based therapies. This antigenic stability suggests that monoclonal antibodies targeting HEV may effectively neutralize the virus without the complications associated with extensive variability present in other viral glycoproteins (e.g., hepatitis C virus E1/E2 or human immunodeficiency virus gp160). As the exact mechanism by which quasi-enveloped HEV particles are neutralized remains enigmatic to date[48,51], the availability of antibodies that either neutralize or do not neutralize quasi-enveloped particles will allow to further investigate individual neutralization mechanisms.

The use of a glycosylated alongside a non-glycosylated P domain facilitated the identification of a glycan-sensitive class of antibodies, targeting the antigenic site termed C3[30], which comprises both weakly neutralizing Abs – in line with a recent report describing the B cell response of vaccinated individuals[31] – and ultrapotent nAbs that have not been described to date. Whether the functional difference between weakly and potent neutralizing Abs targeting C3 can be attributed to the respective contact residues, binding affinity, or yet unknown factors, remains elusive to date. Although the observed interaction of glycan-insensitive antibodies with the dimeric pORF2 decoy suggests a better performance of glycan-sensitive antibodies in infected patients, further studies will be required to directly compare the protective efficacy of different antibodies in vivo. Our results

suggest that glycan-sensitive nAbs constitute approximately 20% of the observed P domain-reactive antibodies, i.e., they are relatively abundant in infected patients that cleared an acute HEV infection, similar to the frequency observed for individuals vaccinated with Hecolin[30]. Additional future studies will be required to understand whether such antibodies might be crucial for the clinical course of HEV infection.

We delineate a rational approach to circumvent the viral escape mechanisms of HEV by leveraging glycan-sensitive nAbs. These antibodies exhibit the capacity to evade the secreted, highly abundant viral antigens that circulate during infection. Our experimental findings are supported by analyses of patient samples from individuals, who have recently cleared an HEV infection, as shown by their negative HEV RNA results. In these samples, "glycan-insensitive" antibodies successfully detected viral antigens in sera that had been treated with detergents to remove the quasi-envelope. In contrast, glycan-sensitive antibodies did not show any reactivity, indicating that they only detect pORF2 in form of infectious virions. This evidence underscores the potential utility of glycan-sensitive nAbs in enhancing therapeutic strategies against HEV infection. It also highlights that our set of well-characterized high-affinity antibodies represents a diagnostic tool that enables us to distinguish glycosylated decoy antigen from infectious particle-associated pORF2, which may provide important insights into the kinetics of pORF2 secretion in acute and chronic HEV infection (as demonstrated in Fig. S9 and Table S7).

## Methods
### Ethical statement

This study was conducted in accordance with the Declaration of Helsinki and is approved by the Ethics committee of the Hannover Medical School (No. 8743_BO_K_2019). All participating individuals (2 individuals for memory B-cell sequencing, 14 individuals for serological analysis) gave written informed consent to the study, utilization of the biomaterial and information on the course of HEV infection and they were not compensated for participation. The patient's sex was determined based on self-report and not considered in the selection of patients. While viral genotyping was not performed for any patient in this study, due to the geographic distribution of HEV in Germany, all patients are assumed to have been infected with HEV GT3. Characteristics of all patients and samples included in this study are provided in Tables S3 and S7. All samples were obtained during routine blood draws and hospital visits. Patients underwent no additional procedures for this study. The work with iPSCs was approved by the Ethics Committee of Heidelberg University (S-439/2018). The HEV GT3 animal experiments performed in Ghent were in accordance with the European Communities Council Directive (86/609/EEC) and approved by the Animal Ethics Committee of the Faculty of Medicine and Health Sciences, Ghent University (refs. ECD17-93 and ECD19-74). All HEV GT1 animal experiments were conducted in accordance with the European Communities Council Directive (86/609/EEC) and were approved by the City of Hamburg, Germany.

### Expression and purification of soluble HEV P domains in Drosophila cells

All genes encoding the HEV pORF2 P domain residues 456–660 (see Table S6 for accession numbers) were cloned into a pMT vector for expression in *Drosophila* S2 cells as described previously[52]. All constructs carried an N-terminal double strep tag, followed by a TEV protease and enterokinase cleavage sites. To identify broadly neutralizing antibodies targeting the infectious particle from HEV GT3, constructs were designed with and without a BiP signal sequence to represent the glycosylated (secreted) and non-glycosylated (non-secreted) form of the P domain, respectively (see list of oligonucleotides in Table S11). A similar strategy was used to clone the non-glycosylated form of P domains from the other human HEV GT and a

rat HEV. Furthermore, proteins used in B-cell isolation and sequencing carried a fluorescent protein (mRuby and mNeon-green for the HEV GT3 glycosylated and non-glycosylated P domains, respectively) between the double strep-tag and TEV cleavage site. A stable S2 cell transfectant was established per construct and the proteins produced as previously described[53] with minor method modifications. Briefly, a total of 2 µg plasmid (1:1 ratio) was co-transfected with 0.1 µg pCoPuro plasmid[54]. Following a 6-day selection period, stable cell lines were then adapted to insect-Xpress media (Lonza). For large-scale production, cells were induced with 4 µM $CdCl_2$ at a density of $6 \times 10^6$ cells/ml. On day 5 post-induction, the non-glycosylated and glycosylated P domains were affinity purified from cell lysates and culture supernatants, respectively, using a Strep-Tactin XT 4Flow column (IBA Lifesciences) followed by size exclusion chromatography on a HiLoad 26/600 superdex 200 pg column (Cytiva) equilibrated in 20 mM HEPES pH 7.4 and 150 mM NaCl. Purified proteins were concentrated and stored at –80 °C.

## B-cell sorting and single-cell sequencing

Both B-cell donors (male patients p60 and p61) presented initially with unknown hepatitis with severely elevated liver enzymes but normal liver function. Patient p60 also presented with mild jaundice. Patient p61 had a mild pre-existing hepatic steatosis whereas p60 had no relevant pre-existing conditions. They were 78 and 45 years old, respectively, at the time of diagnosis, and were both briefly kept for observation in the clinic, where in both cases, HEV RNA led to the diagnosis of acute clinically apparent HEV infection. Frozen PBMCs from the two donors were thawed on ice and resuspended in MACS buffer (PBS pH 7.2, 2 mM EDTA, 0.5% (w/v) BSA) after written consent by the participants. Cells were pelleted ($350 \times g$, 4 °C, 10 min) and resuspended in 80 µl buffer and 20 µl human CD19 microbeads (Miltenyi Biotec) per $10^7$ cells. After a 15-min incubation at 4 °C, cells were pelleted, followed by resuspension of up to $10^8$ cells in 500 µl MACS buffer for magnetic cell separation on LS columns (Miltenyi Biotec) per the manufacturer's instructions. The MACS-sorted cells were labeled (0.2 mg/ml GT3-mNeon-fused P domain (non-glycosylated), 0.2 mg/ml GT3-mRuby-fused P domain (glycosylated), 20 µl APC Mouse Anti Human IgG (BD Bioscience), and 5 µL Alexa Fluor 700 Mouse Anti-Human CD20 (BD Biosciences) per $10^6$ cells in 100 µL, in addition to LIVE/DEAD™ Fixable Near-IR Dead Cell Stain (ThermoFisher Scientific) (1 µl stain in 1 ml)), incubated for 30 min on ice, and later washed with MACS buffer prior to resuspension in 400 µl PBS supplemented with 0.5% (w/v) BSA. Cells were sorted on a BD Bioscience FACSAria III Fusion sorter, and the Chromium Next GEM Single Cell V(D)J Reagent Kits v1.1 was used to process the single-cell solutions. Sequencing of the scBCR libraries was performed on the Illumina NextSeq550 platform using the NextSeq 500 Mid Output Kit v2.5. The flow cytometry data was analyzed using the FCS Express 7 software, while the sequencing data was analyzed with the Loupe V(D)J Browser (Version 3.0.0) and productive sequences were re-annotated with IMGT/HighV-Quest.

## Expression and purification of antibody fragments (scFv and Fab)

Eighty-five antibodies, 48 from donor p60 and 37 from donor p61, were selected for expression in Drosophila S2 cells. Sequences were selected to represent a wide range of heavy-chain germline genes based on clonal abundance, heavy-chain CDRH3 amino acid lengths (5-26 amino acids), according to IMGT numbering, as well as the number of somatic hypermutations (2–37 nucleotides). Codon-optimized synthetic genes comprising the selected scFv sequences were purchased from Twist Biosciences and cloned into a suitable S2 expression vector[34]. Constructs carried an enterokinase (EK) cleavage site and a double strep tag at the C-terminus of the scFv or the C-terminus of the Fab heavy chain. Expression and purification of all constructs

followed the S2 cell expression protocol described above. Purified proteins were concentrated and stored at –80 °C.

## IgG expression and purification

Paired heavy and light chain variable regions of antibodies were amplified from the respective top neutralizing scFv and cloned as IgG1 into a pcDNA3.1 expression vector under the control of a CMV promoter for expression in HEK ExPi293F cells (ThermoFisher Scientific). The cells were transfected using ExpiFectamine™ 293 transfection reagent (ThermoFisher Scientific) following the manufacturer's recommendation with minor method modifications. Briefly, a plasmid-DNA cocktail containing the heavy and light chain-expression plasmids, and plasmids encoding the large T antigen of the SV40 virus and the cell cycle inhibitors p21 and p27 (Heavy chain + light chain + p21 + p27 + SV40 at ratios of 0.345:0.345:0.05:0.25:0.01, respectively), was prepared in 5 ml Opti-MEM (Gibco) at a concentration of 1 µg per ml of final culture volume. The transfection reagent was diluted in 5 ml Opti-MEM and mixed with the plasmid DNA cocktail after a 5-min incubation at room temperature. The mixture was then incubated for 20 min at room temperature, followed by dropwise addition to the cells. Enhancers were added the next day, and after a 4-day expression period, cells were pelleted and IgG affinity-purified from the supernatant using a Protein G column (Cytiva) followed by size exclusion chromatography on a Superdex 200 increase 10/300 column (Cytiva) equilibrated in PBS. Purified mAbs were concentrated and kept at 4 °C until required for further analysis.

## Crystallization, data collection and data processing

Protein preps intended for crystallization were treated with TEV protease and EKMax enterokinase (Thermo Scientific) to remove the affinity purification tags. The non-glycosylated HEV GT3 P domain: antibody fragment complexes (1:1.2 molar ratio) were kept at 16 °C overnight, purified by size exclusion chromatography on a superdex 200 increase 10/300 GL column (Cytiva) and concentrated to 9.3 mg/ml (scFv-p60.1), 6.7 mg/ml (Fab-p60.12), 7.9 mg/ml (scFv-p61.15), and 8.7 mg/ml (scFv-p60.15). The rat HEV: Fab-p60.12 complex was concentrated to 7.3 mg/ml. All crystals were grown at 293 K using the sitting-drop vapor diffusion method in drops mixed in a 1:1 ratio with reservoir solution. Diffracting quality crystals were obtained in the following conditions; p60.1 (2 M ammonium sulfate and 0.1 M sodium acetate pH 4.6), p60.12 (25% w/v PEG 4000, 0.2 M calcium chloride, and 0.1 M Tris pH 8.5), p60.15 (11% w/v PEG 8000, 0.1 M zinc acetate, and 0.1 MES pH 5.5), p61.15 (20% PEG 3350, 0.1 M Bis-Tris propane pH 8.5, and 0.2 M potassium thiocyanate), and p60.12-rat HEV (20% PEG 3350, 0.2 M di-Ammonium hydrogen phosphate). All crystals were flash-frozen in mother liquor containing 30 % ethylene glycol and diffraction data were collected at P11 (DESY) for p60.1 and p60.15, and PX1 (SLS) for p61.15, p60.12, and p60.12-rat HEV. Data were processed, scaled, and reduced with XDS[55], Pointless[56], and programs from the CCP4 suite[57]. Phaser[58] was used for molecular replacement with PDB 3rkc[27] and PDB 6dsi (PDB DOI: 10.2210/pdb6DSI/pdb) as search models to overcome the phase problem. This was followed by model building using Coot[59] with several rounds of refinement using AutoBuster[60] and validation with MolProbity[61]. Figures were generated using PYMOL (http://www.pymol.org/2/support.html).

## ELISA assay

To screen antibodies for binding to the HEV P domains, nunc 96 well ELISA plates (Thermo Scientific) were coated with 100 ng of the glycosylated or non-glycosylated HEV GT3 P domain per well in PBS overnight at 4 °C. Plates were washed with PBS-T and blocked with a blocking buffer (PBS-T and 5 % skimmed milk) for 2 h at room temperature. The plates were then washed with PBS-T and wells loaded with 50 µl of antibody diluted in blocking buffer at a concentration of 2 µg/ml. Following a 1-h incubation at room temperature, the plates

were washed 4x with PBS-T and incubated for 30 min at room temperature with horseradish peroxidase conjugated goat anti-human antibody diluted 1:20,000 in blocking buffer. The plates were washed and developed by adding TMB substrate (BioLegend). The reaction was stopped by addition of 1 M H$_3$PO$_4$ and absorbance measured at 450 nm with 630 nm as reference using the ELx808 absorbance plate reader (BioTek). Data were analyzed using GraphPad Prism 9.

Fourteen additional sera from HEV infected individuals (7 female patients and 7 male patients) with a mean age of 50 years were selected for capture ELISA with patient sera. The ELISA plates were coated with 125 ng of the depicted antibody in PBS for 1 h at 37 °C. Plates were washed and loaded with horseradish peroxidase-conjugated goat anti-rabbit antibody (Invitrogen, reference: 31460, LOT: WC320195) diluted 1:10,000 in blocking buffer. Following a 30-min incubation at 37 °C, the plates were washed and developed by adding TMB (3,3′,5,5′-Tetra-methylbenzidine, Sigma, reference: T0440-1L, LOT:SLCJ5081). The reaction was stopped by adding 1 M sulfuric acid and optical density measured at 450 nm with 630 nm as reference. Data were analyzed using GraphPad Prism 9. The cut-off was calculated by the mean of three negative sera tested or 0.03 if the average was below that, plus 0.16. The ELISA was considered negative when the signal to cut-off value (S/CO) was below 0.9 and positive when it was above 1.1. HEV Antigen positivity of depicted sera was confirmed by performing WANTAI HEV Antigen ELISA according to the manufacturer's protocol (WANTAI BioPharm, reference: #WE-7596). HEV-RNA positivity was tested by the central laboratory at Hannover Medical School using PCR and the RealStar® HEV RT-PCR Kit 2.0 (#272013) from Altona Diagnostics according to the manufacturers protocol.

### Serum depletion assay

Sera from 3 different patients (two female patients and one male patient, aged 46, 59 and 63), recently recovered from an acute HEV infection, diluted 1:50 were incubated with strep-tagged glycosylated and non-glycosylated GT3 P domains (each 250 µg) for 30 min at room temperature. The mixture was then added to MagStrep XT beads (IBA Lifesciences) and incubated for an additional 30 min, with vortexing every 2 min, at room temperature following the manufacturer's instructions. The control sera, diluted 1:100, was incubated with the beads in the absence of either protein. Following incubation, the tubes were transferred to a magnetic rack and supernatant carefully aspirated into a fresh tube. An ELISA assay was used to quantify the depletion. Nunc 96 well ELISA plates were coated overnight with non-glycosylated HEV GT3 P domain as described above. The samples were diluted 1:10 in blocking buffer prior to addition on the ELISA plate and the ELISA assay was performed as described above. To estimate the abundance of glycan-sensitive antibodies, the OD from the un-depleted sample was used to represent total P domain-specific antibodies and was set to 100%. From this, the difference obtained after depletion with the P domains was used to calculate the percentage of glycan-sensitive antibodies present in the sera.

### Surface plasmon resonance

To determine the affinity constants for the HEV P domain interactions with the top neutralizing mAbs, multi-cycle kinetics were performed with the Biacore X100 (Cytiva). All experiments were performed at 25 °C with HBS-EP (0.01 M HEPES pH 7.5, 0.15 M NaCl, 3 mM EDTA, 0.05 % Tween20) as the running buffer. On both flow cells of a CM5 chip (Cytiva) about 2,500 response units (R.U.) StrepTactinXT was immobilized via amine coupling (Twin-Strep-tag Capture kit from IBA Lifesciences). Double-strep-tagged HEV proteins were captured on flow cell 2 and a dilution series of scFvs was injected into both flow cells for 120 seconds at a flow rate of 30 µl/min. After 800 s dissociation, the StrepTactinXT surface was regenerated by injecting 3 M GuHCl for 30 s. All analyses were performed with the sensorgrams of flow cell 2–1 using the Biacore X100 Evaluation software. To determine the kinetic

parameters, the data were fitted with the kinetic setting using a 1:1 binding model and local Rmax. Concentrations, whose Rmax value differed more than one standard deviation from the average Rmax value were excluded from the final analysis.

### Virus production and neutralization assays

Virus production of the HEV GT3 Kernow-C1 p6 G1634R as well as the GT3 HEV 83-2-27 clone was performed as previously described[35]. Briefly, 9 × 10$^6$ cells of the human hepatoma cell line HepG2 were transfected with 7 µg of the viral RNA by electroporation and virus stocks were harvested 5 days later. Cell lysates of transfected cells, generated by three freeze and thaw cycles using liquid nitrogen, were used as naked viral stocks while the 0.45 µm filtered cell culture supernatant served as quasi-enveloped virus stock. Different densities of the vial particles were confirmed by iodixanol gradient and virus stocks were only frozen once at −80 °C until usage. The indicated virus was incubated with different concentrations of mAbs, scFvs or PBS for 1 h at room temperature before infecting HepG2/C3a cells. As a positive control, an anti-HEV IgG positive human serum (dilution 1:100 in cell culture medium) was used. 24 h post infection, fresh medium was added to the cells. After four days, cells were fixed with paraformaldehyde and treated with 0.2 % Triton X-100 solution for 5 min at room temperature for permeabilization. Subsequently, samples were washed and stained overnight with an anti-ORF2 rabbit polyclonal antibody. Infected cells were visualised following staining with Alexa Fluor 488-labeled goat anti-rabbit antibody (Invitrogen, reference: A11008), and focus forming units (FFU) counted using the Elispot CTL system (Immunspot Ultimate S6 UV Imager, Cellular Technology Limited). The half-maximal inhibitory concentration (IC$_{50}$) was determined by titrating mAbs, and dose-response curves were fitted using a non-linear regression model from GraphPad prism 9.

For pre-incubation assays with secreted pORF2, 20 µl of antibody solution with a concentration of 0.8 µg/ml were pre-incubated with 20 µl of the secreted pORF2 dilution at indicated concentrations for 1 h at room temperature. 40 µl of diluted virus stock were added and incubated another hour following addition of 70 µl of the mixture onto HepG2C3a cells. Analysis was performed as described above. As primary antibody for immunofluorescent staining, 1 µg/ml of p60.1 was used to avoid background staining of secreted pORF2 dimers. Alexa Fluor™ 488, goat anti-human antibody (Invitrogen, reference: A-11013) was used as secondary antibody at a dilution of 1:1,000.

### Immunofluorescence

HepG2 cells were transfected with either the full length Kernow-C1 p6 G1634R genome or a subgenomic replicon in which ORF-2 is exchanged by Gaussia Luciferase (GLUC) using electroporation. Subsequently, 20,000 cells per well were seated on 96-well plates. After four days, cells were fixed using 4% paraformaldehyde PBS solution following permeabilization with 0.2% Triton X for 5 min. Cells were incubated with PBS supplemented with 5% horse serum (HS) for 2 h. Selected antibodies in a concentration of 0.25 µg/ml in the same solution were incubated with the cells overnight at room temperature. The next day, plates were washed three times using PBS and Alexa Fluor 488 labeled goat anti-human antibody (see above) was added for another 2 h at a dilution of 1:100 in PBS containing 5% HS. Later, cells were again washed 3 times in PBS. Nuclei were stained with DAPI solution (1:10,000 dilution) for 1 min and washed with ddH$_2$O. Images were acquired using the Olympus IX81 motorized inverted research microscope at ×20 magnification.

### Infection of humanized mice with HEVcc

For infection of human liver-chimeric mice homozygous uPA + /+/SCID mice (species: CB-17/Icr-Prkdcscid/scid/Rj-tg(Alb-uPA$^{+/+}$)) of both sexes were transplanted with approximately 10$^6$ primary human hepatocytes (donor HH342, Corning, The Netherlands) at an age of approximately

2–5 weeks, as described previously (16). The registration number of the used animal facility at the Ghent University is LA1400070. To assess successful liver engraftment, the human albumin concentration in mouse plasma was determined by ELISA (Bethyl Laboratories, Montgomery, Texas, USA). At an age of 3–4 months, mice were then injected with p60.1 ($n = 3$, 1 mg/mouse) or PBS used as control ($n = 3$) on day −1 prior to inoculation with lysate-purified (i.e., naked) cell culture produced HEV p6_wt ($1.01 \times 10^4$ FFU/mouse) on day 0 and a second dose of antibody (1 mg/mouse) or PBS on day 4 post infection given via the intraperitoneal route. Fecal samples and plasma samples were collected on a weekly basis. HEV RNA levels in 10% (w/v) fecal suspensions were quantified using a previously described RT-qPCR protocol[43].

For the co-housing setting, in total 14 female human liver-chimeric homozygous uPA$^{+/+}$/SCID/IL2Rko (USG) mice at an age of approximately 20 weeks were used. The five donor mice were infected via a retrobulbar inoculation with 50–100 µl of a purified stool suspension of a HEV positive patient. Viral titers were monitored in the feces for several weeks. Upon achievement of a stable infection approximately 2–3 weeks after inoculation, blood was taken retrobulbarily from all mice to determine the baseline viral titer levels. For antibody treatment, uninfected mice were injected with p60.1 ($n = 2$, 1 mg/mouse) or a control antibody ($n = 3$) on day -1 prior to establishment of 3 cages with one donor mouse, one p60.1 antibody and one control antibody treated mouse each and two additional doses of antibody (0.5 mg/mouse) on day three and day seven. Co-housing was continued for 12 days and subsequently mice were separated for 6–7 weeks to avoid further uptake of high titer feces from co-housed HEV stably infected mice.

HEV pORF3 was visualized on paraformaldehyde-fixed, cryopreserved liver sections by using a HEV pORF3 specific antibody (1:1600, Bioss, Woburn, USA). Human hepatocytes were co-stained with anti Krt18 antibody (1:400, Santa Cruz, Heidelberg, Germany), which does not react with the murine cells. Specific signals were visualized using an Alexa 555-labeled secondary antibody (1:1200, Invitrogen, Carlsbad, CA, USA) or the TSA Fluorescein System (Perkin Elmer, Jügesheim, Germany) for the amplification of the pORF3 signal.

### Generation of human pluripotent stem cell-derived hepatocyte-like cells (HLCs) and infection with HEV patient isolates

Human induced pluripotent stem cells (iPSCs) from a male donor were differentiated into hepatocyte-like cells (HLCs) in 24-well format as described previously[40]. Fecal samples from 5 HEV patients (3 male patients and 2 female patients, age 21-52) collected at Hannover Medical School were resuspended in PBS, differentially centrifuged, and passed through 0.22 µm filters. For the neutralization assay, the p60.1 bnAb was diluted in hepatocyte culture medium (HCM, Lonza) and incubated with 20 µl of the viral suspension for 1 h at 37 °C on a rolling platform. On day 21 of the differentiation protocol, HLCs were inoculated with the viral suspension with or without bnAB. The inoculum was removed and replenished with fresh HCM 24 h later. 7 days post-infection, HLCs were fixed with 4% PFA, permeabilized and blocked with 10% goat serum, 1% bovine serum, 0.1% Triton X-100 and stained against HEV capsid protein pORF2 using an anti-ORF2 rabbit polyclonal antibody. Stained cells were imaged on the Celldiscoverer7 (Zeiss) and analyzed using Zen software.

### Reporting summary

Further information on research design is available in the Nature Portfolio Reporting Summary linked to this article.

## Data availability

The atomic coordinates and structure factors for five crystal structures were deposited in the Protein Data Bank (http://www.pdb.org/) under the accession codes 8PMW, 8PMX, 8PMY, 8PMZ, and 8PN0. The scRNAseq data were deposited at the European Nucleotide Archive and the Sequence Read Archive under the accession codes PRJEB76306 and PRJNA1166121, respectively. All other raw data generated in this study are provided in the supplementary information or the Source Data file. Source data are provided with this paper.

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

## Acknowledgements

We are grateful to the patients donating samples, to Suzanne Emerson (NIAID, Bethesda, USA) for the hepatitis E virus p6 clone, to Tassilo Volz for generating the humanized USG mice (UKE, Hamburg), and to Marija Backovic (Institut Pasteur, Paris, France) for critical reading of the manuscript. We thank Daniel Depledge (Hannover Medical School, Germany) for helpful discussions. K.D. was supported by a Clinical Leave stipend of the German Center for Infection Research (DZIF, TI 07.001_005). P.B., T.K., M.D. and V.L.D.T are funded by the German Center for Infection Research (DZIF TI 05.714, TTU 05.823 and TTU 05.917); P.B and T.K. also by the Volkswagen-Foundation (9A888). A.V.B., T.K., T.P. and H.W. are funded by the Deutsche Forschungsgemeinschaft (DFG, German Research Foundation) under Germany's Excellence Strategy - EXC 2155 - project number 390874280. G.S. and C.J. were funded by the Deutsche Forschungsgemeinschaft (DFG, German Research Foundation) project 405772731 (A.V.B. and T.K.) and L.J.S. was funded by the Deutsche Forschungsgemeinschaft (DFG,

German Research Foundation) within project B10 of the Collaborative Research Center SFB900 — Projektnummer 158989968 (T.K.). N.P. was funded by the Deutsche Forschungsgemeinschaft (DFG, German Research Foundation) project 500627539 to A.V.B. and T.K. T.K. is funded by the Deutsche Forschungsgemeinschaft (DFG, German Research Foundation) – Projektnummer 439226007. E.S. is funded by the Deutsche Forschungsgemeinschaft (DFG, German Research Foundation) - project number 499292334. A.M., S.P., and V.L.D.T are funded by the Deutsche Forschungsgemeinschaft (DFG, German Research Foundation) – project 272983813. P.M. was supported by the Research Foundation-Flanders (FWO-Vlaanderen) Excellence of Science project VirEOS2.0 and Research Project G0A7Y24N.

## Author contributions

T.K. and P.B. conceptualized and designed the study. G.S., K.D., L.J.S., F.H., L.C., L.H., E.M.G., P.L.N., N.P., L.F., J.G., N.M., S.P., A.-K.M., L.V. and C.J. performed the experiments. G.S., K.D., L.J.S., F.H., L.C., N.P., S.P., A.-K.M. and C.J. analysed the data. E.S., H.W., A.V.B., V.L.D.T, A.K., T.P., M.L., P.M., M.D., T.K., and P.B. interpreted the results. T.K. wrote the initial draft of the manuscript, G.S., K.D., and P.B. revised the manuscript, and E.S., M.D. and all other authors commented on the manuscript.

## Funding

## Competing interests

G.S., K.D., T.K., and P.B. are listed as inventors on a patent "BROADLY NEUTRALIZING ANTIBODIES AGAINST HEPATITIS E VIRUS" describing the use of glycan-sensitive antibodies p60.1 and p60.12 targeting HEV pORF2 for diagnostics, prevention and treatment of HEV infection (EP 22 162 453.9). These authors declare no restrictions on the publication of data and the remaining authors declare no competing interests.

## Additional information

[1]Center of Structural and Cell Biology in Medicine, Institute of Biochemistry, University of Luebeck, Luebeck, Germany. [2]TWINCORE, Centre for Experimental and Clinical Infection Research, A Joint Venture between Helmholtz-Centre for Infection Research and Hannover Medical School, Hannover, Germany. [3]Department of Gastroenterology, Hepatology, Infectious diseases and Endocrinology, Hannover Medical School, Hannover, Germany. [4]German Center for Infection Research (DZIF), Braunschweig, Germany. [5]Institute of Virology, Hannover Medical School, Hannover, Germany. [6]Department of Internal Medicine, University Medical Center Hamburg-Eppendorf, Hamburg, Germany. [7]Laboratory of Liver Infectious Diseases, Department of Diagnostic Sciences, Faculty of Medicine and Health Sciences, Ghent University, Ghent, Belgium. [8]Schaller Research Group, Department of Infectious Diseases, Virology, University Hospital Heidelberg, Center for Integrative Infectious Diseases Research (CIID), 61920 Heidelberg, Germany. [9]Centre for Individualised Infection Medicine (CiiM), a joint venture between Helmholtz-Centre for Infection Research and Hannover Medical School, Hannover, Germany. [10]Department of Molecular and Medical Virology, Ruhr University Bochum, Bochum, Germany. [11]Cluster of Excellence RESIST (EXC 2155), Hannover Medical School, Hannover, Germany. [12]University Medical Center Hamburg-Eppendorf, Institute of Medical Microbiology, Virology and Hygiene, Hamburg, Germany. [13]Centre for Structural Systems Biology (CSSB), Hamburg, Germany. [14]These authors contributed equally: George Ssebyatika, Katja Dinkelborg. ✉e-mail: thomas.krey@uni-luebeck.de; patrick.behrendt@twincore.de

