## [Transparent Peer Review file · Nature Communications]

Broadly neutralizing antibodies isolated from HEV convalescents confer protective effects in human liver-chimeric mice

Corresponding Author: Professor Thomas Krey

Editorial Note: Parts of this peer review file have been redacted as indicated to maintain the confidentiality unpublished data.

Version 0:

Reviewer comments:

Reviewer #1

(Remarks to the Author)

HEV produces two different forms of ORF2 proteins (pORF2): one makes the capsid and another is secreted from infected cells and functions as a decoy that competes for capsid-specific antibodies. Unlike the capsid, the decoy is glycosylated at the apex of the protruding (P) domain masking certain epitopes on the capsid. In this study, Ssebyatika et al. identified glycan-sensitive and glycan-insensitive HEV-specific monoclonal antibodies (mAbs) from hep E patients. The glycan-sensitive mAbs bind to the virus particles but not the secreted glycosylated ORF2 dimers, whereas the glycan-insensitive ones bind to both. They show that HEV neutralization by glycan-sensitive mAbs was resistant to the decoy in an in vitro neutralization assay. Furthermore, they solved the structures of two glycan-sensitive and two glycan-insensitive mAbs in complex with the P domain. Lastly, they show that administration of a glycan-sensitive mAb P60.1 protected against HEV infection in humanized mice.

Overall, this is a well written paper and the findings are novel. The glycan-sensitive antibodies identified in this study provide a new tool to distinguish virus particles from the decoy antigen and have a potential therapeutic value. The methodology is sound, and the data are clearly presented and of high quality. However, there is a lack of direct comparison between the glycan-sensitive and glycan-insensitive antibodies in some experiments which seems important to do and would strengthen authors' conclusion.

Specific points.

Lines 154-155 and line 428: the numbering of amino acids in the P domain dimer (aa 456-660) is inconsistent with the numbering in Fig. S1 (aa 456-606). Please clarify.

Some of the figures were incorrectly referred to in the main text. For example, on line 165, Fig. 1b should be Fig. 1c. On line 170, Fig. 1c-e should be Fig. 1b. And on line 172, Fig. 1c should be Fig. 1b.

Fig. 1i shows that some of the human mAbs neutralized quasi-enveloped HEV. However, work from Okamoto and colleagues shows that quasi-enveloped HEV are resistant to neutralizing anti-capsid antibodies. Please discuss the possible reasons for these different results.

Fig. 2c and lines 256-257: the result that all four potent nAbs recognized pORF2 in sera seems to conflict with the results in Fig. 2a showing that p60.1 and p60.12 did not bind to the glycosylated P domain.

Fig. 2d shows neutralization of clinical isolates by the glycan-sensitive nAb p60.1. Since the results in Fig. 2b show that glycan-sensitive nAbs are resistant the decoy but the glycan-insensitive ones are not, a reasonable experiment will be comparing p60.1 with the glycan-sensitive nAbs side-by-side to see if p60.1 neutralizes serum virus more efficiently.

Fig. 5 shows P60.1 protects against HEV intraperitoneal infection and co-housing transmission. While the result is nice, it does not mean that the glycan-sensitive antibodies are better than existing antibodies. It would be interesting to test whether this antibody performs better than the glycan-insensitive antibodies in the presence of the decoy in vivo. For example, one can inject these antibodies to animals with already established infection to see if any of these antibodies reduces the virus titer.

Fig. S6: It states that approximately 20% of anti-P domain antibodies were not depleted by glycosylated P domain. How was this 20% calculated? It would also be interesting to assess the proportion of glycan-sensitive antibodies in sera from chronically infected patients.

Lines 411-414: Since most pORF2 in patient' serum are glycosylated and the infectious particles-associated pORF2 are difficult to detect, it is not clear what the authors meant by stating "the glycan-sensitive antibodies will be an important diagnostic tool". Also, since the glycan-sensitive antibodies recognize nonglycosylated capsid, it is unclear these antibodies will "provide insights into the kinetics of pORF2 secretion and the role of the secreted pORF2 dimer".

Reviewer #2

(Remarks to the Author)

Thank you for allowing me to review the article titled 'Human "glycan-sensitive" antibodies protect from hepatitis E virus infection'. Hepatitis E virus (HEV) causes 3.3 million symptomatic infections and 44,000 deaths per year. Individuals with low immune function may develop chronic infections, and pregnant women may suffer from fulminant disease due to HEV infection. Despite these important public health implications, no specific antiviral treatment has been approved to date. Currently, there is a highly effective HEV vaccine based on recombinant virus-like particles, which consists of a truncated version of the capsid protein pORF2, and can induce protective humoral immune responses to prevent the symptomatic occurrence of GT4 infection. However, this vaccine has only been licensed in China and Pakistan. Global promotion of the HEV vaccines may be an urgent issue to be addressed in the field of HEV treatment/prevention. Ribavirin has been proven to be effective in treating HEV infection. Based on the above information, the significance of developing therapeutic drugs based on neutralizing antibodies in this study is generally modest. There are still some issues in this study that need further improvement:

1. The lack of flow cytometry results in this study makes it impossible to determine the reliability of the information described, such as the proportion of specific B cells.
2. Please specify the basis for selecting these 85 antibodies for recombinant expression and subsequent property testing. (Line: 178-180)
3. Please show information of these antibodies in detail, such as by table.
4. Should test binding activities of the 85 antibodies, before neutralization assay.
5. Whether all 3,364 antibodies are confirmed as HEV-specific antibodies through binding activity testing. If not, then conducting genetic feature analysis on non-specific antibodies is meaningless for exploring the response pattern of HEV antibodies, and can instead mislead readers. (Line: 158-162; 166-177)
6. The author used two types of P domain proteins for specific antibody screening, but some antibodies only bind to the non-glycosylated P domain. Is there a lack of strict gate selection during sorting? (Figure 2a)
7. Lack of sufficient antibody to support this conclusion. (Line: 211-212)
8. It is necessary to show the kinetic curve of affinity detection. (Table S1)
9. The authors need to evaluate the therapeutic and preventive effects of antibodies separately. (Figure 5a)
10. The author should add a set of animal experiment to continuously monitor the preventive effect of antibodies on co-housing infection for more than 13 days.
11. The crystal structure analysis of the complex should detail and display the key amino acid information of antigen-antibody interactions. The authors should focus on analyzing the conserved characteristics of these key amino acids. The conserved characteristics of the epitope recognized by P60.12 should be fully demonstrated in the text and main figure. (Figure 3-4)

Reviewer #3

(Remarks to the Author)

In this study, Ssebyatika et al. provide new insights into 'glycan-sensitive' human antibodies against Hepatitis E virus (HV) using the classic technique of antigen-specific single-cell sorting. The focus of their investigation is on the unique properties of pORF2 dimers, which are believed to act as decoys for neutralizing antibodies. These dimers exhibit distinct glycosylation patterns compared to pORF2 found in infectious particles, prompting the researchers to emphasize the significance of 'glycan-sensitive' antibodies. They then used single cell sequencing to identify monoclonal antibodies (mAbs) and finally focused on p60.1 and p60.12 as "glycan-sensitive" antibodies, and p60.15 and p61.15 as "glycan-insensitive" antibodies. While previous studies have already recognized the importance of pORF2 as an antigen, this manuscript suggests that p60.1 could also be employed as a therapeutic and diagnostic tool to determine whether an individual is infected with the HEV virus or not.

The research design is appropriate, and the conclusions are supported by the experimental results. Overall, the manuscript is well-written. I have only a few suggestions for the authors.

Major Comments:

1. The section about the sorting and single cell sequencing is not very well developed:
 - I. Representative flow/sorting gates should be shown.
 - II. It is not clear which cells were sorted: from the text it appears that only double positive were sorted (line 159-160) but then many of the cloned scFv recognized only the non-glycosylated P domain (Fig 2a).
 - III. Authors discuss that few clonally related sequences were found per individual (line 163): some basic B cell analysis

showing mutational load, clonality, etc should be shown.

IV. Upon publication the scRNAseq data should be deposited.

2. The number of experiments and mice for each experiment should be reported. It appears that the in vivo experiments were performed only once with 3 mice per group. If that is the case they should be repeated.

3. Regarding the protection experiment: it is interesting and concerning that one mouse fails to clear the virus. In line 381 you suggest that p60.1 protects from HEV challenge via two different routes, however 1/3 mice (33%) has a viral titer after few weeks so the virus is not controlled. This should be discussed more extensively, including possible reasons and implications

4. If possible, it is advisable to include results from animal experiments infected with different subtypes of HEV

5. The motivation for choosing pORF2 as the target is not sufficiently strong, especially for not expert in the field and considering that Nature Communications is a multidisciplinary journal. The introduction clarifies the HEV infection process and the multiple functions of antibodies. Could you add some reasons and advantages to the selection of pORF2 as the antigen of interest?

6. Regarding epitope specificity, it would be nice to have an alignment with highlighted p60.1-binding site conservation among different HEV genotypes.

Minor Comments:

1. Line 165: the correct call is Fig 1c

2. Line 170 the correct call is Fig 1b-e

3. Line 279: "was in the low nanomolar range (Table S3)" according to the table the affinity is in the micromolar range.

4. Figure S3: The concentration of IgG here is 10µg/mL, which is different from Figure 2b. Please comment on this.

Reviewer #4

(Remarks to the Author)

This paper by Ssebyatika et al., describes novel glycan sensitive antibodies that protect against Hepatitis E virus infection. They used antigen-specific single-cell sorting, next generation sequencing, structural biology, and functional and biochemical antibody characterization in vitro and in vivo to understand the antibody response induced by acute HEV infection in humans.

Their results can have implications for the design of a cross-protective HEV vaccine, and highlight the potential of "glycan-sensitive" human antibodies as novel tool to prevent and possibly treat HEV infection.

Overall the paper is clear and well written although some clarifications could be made. The paper should be of interest to readers in the field of virology, antibody, and vaccine design.

For the non Hepatitis E virus experts, it would be helpful if the authors explain how the infectious particles do not have glycans that are present at some other stages and the differences between the naked vs the quasi-enveloped virions. It may be helpful to have schematic showing the different compartments and the different glycosylation sites as well as the known antigenic sites.

Need to clarify the signal peptide -it seems that the construct 1 shown in Figure S1 does not have a signal peptide but how is this protein expressed in S2 cells?

In fig S1, other potential glycosylation sites are shown - are they present? The authors only mention one glycosylation site. Mass spec could be done to confirm.

Does the recombinant HEV vaccine (Hecolin) bind to the glycan sensitive abs described here?

Can the authors resolve glycans or at least proximal NAG in the crystal structure of the glycosylated dimer? This is not mentioned in the text.

Minor comments

Line 70 GT 1 or GT1 -

Would be nice to see B cell sorting plots as supplemental.

Lines 133-134: reads weird

X-ray crystallographic analyses showed that targeted neutralization epitopes involved N562 within a highly conserved region at the pORF2 P domain dimer apex.

Fig 2a. What do the VH labels mean?

Fig3. Label N3 on the figure - can be hard to see otherwise

Only on scFV and one Fab shown on the dimer which is confusing even though the authors write this is done for clarity. Maybe each promoter could be colored in different shades of grey to see better the dimer interface.

Line 210-211:

Rewrite: to "suggesting that this group of antibodies do not share a common genetic signature"

Would be nice to show BLI and SPR curves in supplemental

Lines 232-237: paragraph could be rewritten:

It is unclear if this is done with sera or with monoclonal antibodies. Is the dimer/nAb complex removed prior neutralization? The case for escape mutations should be put into context of variability of this virus (compared to others).

Details on interacting residues are not shown in figures 3 and 4. That could be added.

Version 1:

Reviewer comments:

Reviewer #1

(Remarks to the Author)

The revised manuscript has been improved. However, several issues remain.

1. Lines 162 and 475: authors state these numbers (45-660) are correct. However, according to the information in Fig. S1, the P domain used for B cell was C terminally truncated (456-606). Please double check.
2. Lines 221-222: "highlighting that such a 'glycan-sensitive' antibody response can be induced in most patients". This may be overstated. Only two patients were analyzed in this study and there was only one such mAb was identified in patient #61.
3. Fig. S5: The neutralization is not efficient for most mAbs including p60.12, p60.15 and p61.15. It would be hard to claim that these mAbs are superior to existing mAbs.
4. Fig. 5a, lines 380-382: "...the kinetics of the increase in viral RNA correlated with the decrease in antibody titer, suggesting that the presence of sufficient bnAb quantities in the serum can control HEV infection and minimize virus shedding". This explanation is not satisfactory. All 3 animals in the antibody treated group received the same quantities of antibody and virus. The decline of antibody in the serum is not sufficient to explain why 1 out of 3 animals got infected especially when only the data from one animal is shown.
5. Lines 417-419: It has been shown that quasi-enveloped particles are completely cloaked in host membranes. Epitope specificity and kinetic binding do not seem to provide a satisfactory explanation for the efficient neutralization of quasi-enveloped particles.
6. Lines 421-422: "The efficient block of liver infection in mice pretreated with p60.1 suggests efficient neutralization of quasi-enveloped particles". Please clarify if the inoculum used for the animal experiments was naked or quasi-enveloped. If naked virus was used, the conclusion would not be convincing.
7. Lines 453-454: Does it mean that glycan sensitive Ab recognize eHEV in the blood?
8. Since the whole idea of this paper is that the glycan-sensitive antibody will perform better in patients than glycan-insensitive antibody, I still think that this idea needs to be vigorously tested in vivo where these two types of antibodies are compared in the presence and absence of the decoy.

Reviewer #2

(Remarks to the Author)

The quality for publication has already been met. However, it is still recommended that the author approaches the HEV-specific binding capacity analysis of the 3,364 antibodies. If the specificity is too low, subsequent analyses of the sequence characteristics of so-called specific antibodies could mislead researchers working on HEV-related studies.

Reviewer #3

(Remarks to the Author)

The authors appropriately addressed most of the comments from the reviewers. The paper is a valuable addition to the scientific literature.

I have still few minor comments:

- 1) In new Figure S2 the gate is placed on double antigen negative population. Took me a while to understand that those were the cells not sorted. Would be better to include another gate/shape to highlight what was actually sorted.
- 2) I am still of the opinion that the animal experiments should have been repeated, as these two experiments show different protection mechanisms, but the editor can make the call according to the journal policy. If experiments are not repeated, I think the limitation should be more clearly discussed within the paper.
- 3) New Table S1. The resolution is too poor to read

Reviewer #4

(Remarks to the Author)

The authors have addressed the comments.

Rebuttal letter

We thank all reviewers for their constructive comments and their positive feedback, for which we provide a point-by-point response below:

Reviewer #1: Summary

HEV produces two different forms of ORF2 proteins (pORF2): one makes the capsid, and another is secreted from infected cells and functions as a decoy that competes for capsid-specific antibodies. Unlike the capsid, the decoy is glycosylated at the apex of the protruding (P) domain masking certain epitopes on the capsid. In this study, Ssebyatika et al. identified glycan-sensitive and glycan-insensitive HEV-specific monoclonal antibodies (mAbs) from hep E patients. The glycan-sensitive mAbs bind to the virus particles but not the secreted glycosylated ORF2 dimers, whereas the glycan-insensitive ones bind to both. They show that HEV neutralization by glycan-sensitive mAbs was resistant to the decoy in an in vitro neutralization assay. Furthermore, they solved the structures of two glycan-sensitive and two glycan-insensitive mAbs in complex with the P domain. Lastly, they show that administration of a glycan-sensitive mAb p60.1 protected against HEV infection in humanized mice.

Overall, this is a well written paper, and the findings are novel. The glycan-sensitive antibodies identified in this study provide a new tool to distinguish virus particles from the decoy antigen and have a potential therapeutic value. The methodology is sound, and the data are clearly presented and of high quality.

We thank the reviewer for the positive feedback.

Comments

Lines 154-155 and line 428: the numbering of amino acids in the P domain dimer (aa 456-660) is inconsistent with the numbering in Fig. S1 (aa 456-606). Please clarify.

We thank the reviewer for pointing out this inconsistency in labelling. The numbering has been rectified to 456-660 in both cases, the shorter amino acid range defines the visible electron density within the structure and was therefore erroneously used at these positions in the text.

Some of the figures were incorrectly referred to in the main text. For example, on line 165, Fig. 1b should be Fig. 1c. On line 170, Fig. 1c-e should be Fig. 1b. And on line 172, Fig. 1c should be Fig. 1b.

We apologize for the error in referencing. We have re-arranged the figure and carefully revised the main text and the figures referenced therein accordingly.

Fig. 1i shows that some of the human mAbs neutralized quasi-enveloped HEV. However, work from Okamoto and colleagues shows that quasi-enveloped HEV are resistant to neutralizing anti-capsid antibodies. Please discuss the possible reasons for these different results.

We thank the reviewer for this important remark and we are aware of the results published by Okamoto et al.. We were thus equally surprised to see the potent neutralization of quasi-enveloped particles.

Gradient ultracentrifugation of the quasi-enveloped and naked virus stocks allowed us to exclude a major contamination of quasi-enveloped particles with naked virus to cause the observed neutralization, as it facilitates a clear separation of viral RNA from the fractions containing lower density particles in the quasi-enveloped virus stock. Although it is difficult to formally exclude minute amounts of naked virions being present, the majority of particles in the supernatant are quasi-enveloped particles of lower density.

Additionally, we demonstrated protection of mice after intravenous treatment with our antibodies against oral HEV challenge (Fig. 5), while untreated or control-treated animals unambiguously showed infection in liver sections, most likely via the blood. As it is currently believed that mostly quasi-enveloped particles circulate in patients, this corroborates the observed neutralization of quasi-enveloped particles by our antibody.

We currently favor the hypothesis that was described by Das et al. (Nat Rev Microbiol, 2023; PMID 37185947) given the very high affinity of the antibodies to their cognate antigen. Hepatitis A virus, exhibiting a similar life cycle as HEV, also circulates as quasi-enveloped virus and is notably inhibited by neutralizing antibodies administered as post-exposure prophylaxis (Victor et al., NEJM, 2008; PMID 17947390). In analogy, the primary mechanism of protection offered by the current HEV vaccine (Hecolin®), is the induction of neutralizing antibodies, suggesting that circulating quasi-enveloped HEV is susceptible to neutralization by antibodies. The differences to the work of Okamoto et al. might be due to antibody

characteristics like epitope specificity or affinity, or to the different cell culture system and detection method used. We have now appended a short paragraph further explaining this open question in the concluding remark section of the revised manuscript (lines 411-416).

Fig. 2c and lines 256-257: the result that all four potent nAbs recognized pORF2 in sera seems to conflict with the results in Fig. 2a showing that p60.1 and p60.12 did not bind to the glycosylated P domain.

We thank the reviewer for pointing out this apparent contradiction. As the sera of the patients selected were positive for HEV RNA, two forms of pORF2 are expected to be present, 1) non-glycosylated pORF2 in form of infectious capsids and 2) glycosylated pORF2 dimers likely serving as antibody decoy.

As described before (Montpellier et al., *Gastroenterology*, 2018; PMID 28958858), the most abundant antigen present in the sera is most likely the glycosylated pORF2 dimer. Since the glycan-sensitive antibodies p60.1 and p60.12 only recognize non-glycosylated pORF2 (i.e., as infectious capsids), but not the glycosylated pORF2 dimer, using these antibodies with individual patient sera is expected to yield lower OD values than using glycan-insensitive antibodies p60.15 and p61.15 as capture reagents.

In the revised manuscript we have now clarified this issue further (lines 261-265)

Fig. 2d shows neutralization of clinical isolates by the glycan-sensitive nAb p60.1. Since the results in Fig. 2b show that glycan-sensitive nAbs are resistant the decoy but the glycan-insensitive ones are not, a reasonable experiment will be comparing p60.1 with the glycan-sensitive nAbs side-by-side to see if p60.1 neutralizes serum virus more efficiently. [redacted]

We thank the reviewer for this insightful comment. It is important to note that the clinical isolates used in Fig. 2d were purified from stool samples using density gradient centrifugation (see methods) and are thus not expected to contain glycosylated pORF2 dimers in a similar way as serum virus. Neutralization assays using serum virus have been reported only once in the literature (Takahashi et al., *J Clin Microbiol*, 2010; PMID 20107086) and not yet been reproduced by other labs likely due to the number of variables including endogenous antibody titers, unspecific inhibition by sera, pORF2 dimer levels, virus titer. Although we have heavily invested in such assays, we currently consider the obtained results not sufficiently reproducible to publish them.

[redacted]

[redacted]

Fig. 5 shows p60.1 protects against HEV intraperitoneal infection and co-housing transmission. While the result is nice, it does not mean that the glycan-sensitive antibodies are better than existing antibodies. It would be interesting to test whether this antibody performs better than the glycan-insensitive antibodies in the presence of the decoy *in vivo*. For example, one can inject these antibodies to animals with already established infection to see if any of these antibodies reduces the virus titer.

We understand the reviewer's concern, and we thank the reviewer for the appreciation of these nice results. We currently do not feel that a direct *in vivo* comparison between two individual antibodies (either glycan-sensitive or glycan-insensitive) is feasible at this point. In this manuscript, we provide first evidence that potent human neutralizing antibodies can protect from HEV infection in human liver-chimeric mice. Although the observed sensitivity of glycan-insensitive antibodies to the dimeric pORF2 decoy suggests a better performance of glycan-sensitive antibodies *in vivo*, further studies will be required to directly compare the protective efficacy of different antibodies. In the revised manuscript we have now clarified this issue further (lines 435-438).

Fig. S6: It states that approximately 20% of anti-P domain antibodies were not depleted by glycosylated P domain. How was this 20% calculated? It would also be interesting to assess the proportion of glycan-sensitive antibodies in sera from chronically infected patients.

To estimate the abundance of glycan-sensitive antibodies, the optical density from the corresponding un-depleted sample (representing the total amount of P domain-specific antibodies) was set to 100%. The difference to this value obtained after depletion with either P domain was used to estimate an approximate percentage of glycan-sensitive antibodies present in the sera. We have further appended this information to the serum depletion assay in the methods section.

We appreciate the reviewer's perspective that this assay enables comparison of glycan-sensitive antibody proportions across different clinical courses of HEV infection, including longitudinal analysis of both acute and chronic phases. However, we believe that conducting such a study necessitates highly defined and well-characterized patient cohorts, which are beyond the current manuscript's scope. Furthermore, such comparisons are complicated by the intricate nature of

antigen-antibody interactions. Statistically and relatively, comparisons between acutely and chronically infected individuals, especially when contrasting immunosuppressed individuals with those who are immunocompetent, may yield significantly different results. This is due to potential variations in antibody titers, affinities, and antigen levels among these groups.

Lines 411-414: Since most pORF2 in patient' serum are glycosylated and the infectious particles-associated pORF2 are difficult to detect, it is not clear what the authors meant by stating "the glycan-sensitive antibodies will be an important diagnostic tool". Also, since the glycan-sensitive antibodies recognize non-glycosylated capsid, it is unclear these antibodies will "provide insights into the kinetics of pORF2 secretion and the role of the secreted pORF2 dimer".

We understand that this paragraph was not unambiguous. We believe that a tool that facilitates a comparative analysis of the kinetics of non-glycosylated pORF2 (associated to infectious particles) and glycosylated pORF2 dimer in the course of infection will represent an important tool to further understand pORF2 biology. We have clarified this paragraph further in the revised manuscript. To further illustrate the diagnostic value of these antibodies, we have appended an additional figure to the revised manuscript that describes two clinical courses of infection longitudinally by different ELISAs and RT-PCR (see Fig. S8 and Table S7 of the revised manuscript).

Reviewer #2: Summary

Thank you for allowing me to review the article titled 'Human "glycan-sensitive" antibodies protect from hepatitis E virus infection'. Hepatitis E virus (HEV) causes 3.3 million symptomatic infections and 44,000 deaths per year. Individuals with low immune function may develop chronic infections, and pregnant women may suffer from fulminant disease due to HEV infection. Despite these important public health implications, no specific antiviral treatment has been approved to date. Currently, there is a highly effective HEV vaccine based on recombinant virus-like particles, which consists of a truncated version of the capsid protein pORF2 and can induce protective humoral immune responses to prevent the symptomatic occurrence of GT4 infection. However, this vaccine has only been licensed in China and Pakistan. Global promotion of the HEV vaccines may be an urgent issue to be addressed in the field of HEV treatment/prevention. Ribavirin has been proven to be effective in treating HEV infection. Based on the above information, the significance of developing therapeutic drugs based on neutralizing antibodies in this study is generally modest. There are still some issues in this study that need further improvement:

We thank the reviewer for the feedback.

Major criticisms

1. The lack of flow cytometry results in this study makes it impossible to determine the reliability of the information described, such as the proportion of specific B cells.

We apologize for omitting these results. We have now appended an additional supplementary figure to the revised manuscript (Figure S2) showing the FACS plots.

2. Please, specify the basis for selecting these 85 antibodies for recombinant expression and subsequent property testing. (Line: 178-180)

The 85 antibody sequences were selected based on clonal abundance, CDRH3 length and the number of somatic hypermutations. We have now appended this information to the paragraph on “Expression and purification of antibody fragments” in the methods section.

3. Please, show information of these antibodies in detail such as by table

We have now added a supplementary table (Table S1) correlating antibody features including the heavy and light chain CDRs and V segments to the neutralization results obtained for scFvs.

4. Should test binding activities of the 85 antibodies, before neutralization assay.

We understand that testing antibody binding first is the standard for the identification of antibodies. However, this approach will identify both neutralizing and non-neutralizing antibodies and a second round of selection using a neutralization assay is required to identify the potent nAbs. As our main goal was to identify potent neutralizing antibodies, we preferred to select antibodies directly based on neutralization potency rather than binding activity.

5. Whether all 3,364 antibodies are confirmed as HEV-specific antibodies through binding activity testing. If not, then conducting genetic feature analysis on non-specific antibodies is meaningless for exploring the response pattern of HEV antibodies and can instead mislead readers. (Line: 158-162; 166-177)

We respectfully disagree with the reviewer on this issue. From the 85 antibodies that we expressed, 52 reduced HEV infection by >20% at 10 µg/ml, indicating a specificity for pORF2 in line with the pORF2-specific sorting. A considerable number of non-neutralizing pORF2-specific antibodies can be assumed based on

other antigen-specific antibody isolation reports, suggesting the vast majority of the 85 selected antibodies to be HEV-specific. As there is no reason to believe that we preferentially chose HEV-specific antibodies during the initial selection process, we therefore estimate that at least 75-85% of antibodies within the entire dataset are HEV-specific. We thus consider the genetic analysis from the entire 3,364 antibodies very meaningful data that should be published, in particular since this is the first dataset of its kind from HEV-infected patients.

6. The author used two types of P domain proteins for specific antibody screening, but some antibodies only bind to the non-glycosylated P domain. Is there a lack of strict gate selection during sorting? (Figure 2a)

We agree with the reviewer that the text on the sorting was not unambiguous. In fact, double-positive cells and single-positive cells in either color were sorted separately and subsequently pooled given the low cell numbers of double positive cells to render single-cell sequencing most efficient. Therefore, both populations (1) Abs recognizing both proteins and (2) Abs recognizing only the non-glycosylated protein were expected to be present. Due to the identical amino acid sequence no antibodies recognizing only glycosylated protein were expected. We have further clarified this issue in the revised manuscript (lines 159-161).

7. Lack of sufficient antibody to support this conclusion. (Line: 211-212)

This sentence has been rephrased into "..., suggesting that this group of antibodies do not share a common genetic signature".

8. It is necessary to show the kinetic curve of affinity detection. (Table S1)

We have now appended an additional supplementary figure (Fig. S4) showing representative chromatograms that served for the determination of kinetic binding parameters.

9. The authors need to evaluate the therapeutic and preventive effects of antibodies separately. (Figure 5a)

We agree with the reviewer that the therapeutic and prophylactic effects of p60.1 need to be addressed separately. To get first insights into the in vivo efficacy of p60.1, we have addressed its prophylactic effect in human-liver chimeric mice in this study. While we will certainly also evaluate the therapeutic effect of this

antibody, we consider such an extensive animal experiment beyond the scope of this manuscript.

10. The author should add a set of animal experiment to continuously monitor the preventive effect of antibodies on co-housing infection for more than 13 days.

We agree with the reviewer that additional analyses to monitor the prophylactic effect of these antibodies in more detail would be interesting to have. However, it is important to note that all animals in the control group got infected during the 13 days of co-housing, whereas none of the antibody-treated animals got infected. Therefore, we consider additional animal experiments at this point not essential to conclude that our antibody protects from HEV infection via co-housing.

11. The crystal structure analysis of the complex should detail and display the key amino acid information of antigen-antibody interactions. The authors should focus on analyzing the conserved characteristics of these key amino acids. The conserved characteristics of the epitope recognized by p60.12 should be fully demonstrated in the text and main figure. (Figure 3-4)

We have appended an alignment (Fig. S10) highlighting the conservation of P domain residues across HEV genotypes (including genotype present in Hecolin) involved in the interaction with the glycan-sensitive antibodies p60.1 and p60.12. We have mentioned these additional figures in the main text and revised the manuscript accordingly.

Reviewer #3: Summary

In this study, Ssebyatika et al. provide new insights into 'glycan-sensitive' human antibodies against Hepatitis E virus (HEV) using the classic technique of antigen-specific single-cell sorting. The focus of their investigation is on the unique properties of pORF2 dimers, which are believed to act as decoys for neutralizing antibodies. These dimers exhibit distinct glycosylation patterns compared to pORF2 found in infectious particles, prompting the researchers to emphasize the significance of 'glycan-sensitive' antibodies. They then used single cell sequencing to identify monoclonal antibodies (mAbs) and finally focused on p60.1 and p60.12 as "glycan-sensitive" antibodies, and p60.15 and p61.15 as "glycan-insensitive" antibodies. While previous studies have already recognized the importance of pORF2 as an antigen, this manuscript suggests that p60.1 could also be employed as a therapeutic and diagnostic tool to determine whether an individual is infected with the HEV virus or not.

The research design is appropriate, and the conclusions are supported by the experimental results. Overall, the manuscript is well written. I have only a few suggestions for the authors.

We thank the reviewer for the positive feedback.

Major comments

1. The section about the sorting and single cell sequencing is not very well developed:
I. Representative flow/sorting gates should be shown.

As outlined in the response to a comment by reviewer #2 (see above) we have now prepared an additional supplementary figure (Fig. S2) showing the requested flow cytometry plots.

II. It is not clear which cells were sorted: from the text it appears that only double positive cells were sorted (line 159-160) but then many of the cloned scFv recognized only the non-glycosylated P domain (Fig 2a).

As outlined in the response to a comment by reviewer #2 (see above) we have now further clarified the text on the sorting process in the revised manuscript.

III. Authors discuss that few clonally related sequences were found per individual (line 163): some basic B cell analysis showing mutational load, clonality, etc. should be shown.

We agree with the reviewer that this is an important piece of information. Fig. 1g shows the mutation rate in the heavy chain variable regions. We have now appended two more panels illustrating the clonality and mutation rate in the light chain variable regions to the revised Fig. 1.

IV. Upon publication, the scRNAseq data should be deposited.

The scRNAseq data has been deposited at the European Nucleotide Archive (PRJEB76306) and the Sequence Read Archive (PRJNA1166121).

2. The number of experiments and mice for each experiment should be reported. It appears that the in vivo experiments were performed only once with 3 mice per group. If that is the case, they should be repeated.

We agree with the reviewer that the number of mice per experiment (3 mice in each experiment) is insufficient to draw firm conclusions based on an individual experiment. However, in this study we performed two independent experiments in two independent labs using two independent challenge viruses (GT1 and GT3,

respectively) to protect human liver transgenic mice from HEV challenge infection via two independent routes of infection and we observed a strong protective effect in both experiments. We consider this consistency in the observed effect strong evidence for its reproducibility and therefore feel that a pure repetition of this experiment in the same animal model is not required to draw the conclusions presented in the manuscript.

3. Regarding the protection experiment: it is interesting and concerning that one mouse fails to clear the virus. In line 381 you suggest that p60.1 protects from HEV challenge via two different routes, however 1/3 mice (33%) have a viral titer after few weeks, so the virus is not controlled. This should be discussed more extensively, including possible reasons and implications.

We understand the reviewer's concern in this issue. We performed very careful Next-Generation Sequencing of HEV isolated from this particular mouse and the sequence analysis revealed no escape mutations in the region encoding the p60.1 epitope. Together with the inverted correlation between the decrease in antibody titer and the increase in viral RNA these results indicate that indeed the antibody is able to control the virus as long as sufficient amounts are circulating. We have now included the information on NGS in the revised manuscript and further clarified this issue (lines 374-376).

4. If possible, it is advisable to include results from animal experiments infected with different subtypes of HEV

We thank the reviewer for this suggestion. This is exactly what we have done, since the inoculation challenge experiment was performed with GT 3 and the co-housing challenge experiment was carried out with GT 1.

5. The motivation for choosing pORF2 as the target is not sufficiently strong, especially for not expert in the field and considering that Nature Communications is a multidisciplinary journal. The introduction clarifies the HEV infection process and the multiple functions of antibodies. Could you add some reasons and advantages to the selection of pORF2 as the antigen of interest?

We have now added a sentence in the introduction of the revised manuscript explaining the choice of antigen better.

6. Regarding epitope specificity, it would be nice to have an alignment with highlighted p60.1-binding site conservation among different HEV genotypes.

As outlined in the response to a comment by reviewer #2 (see above) we have

now appended an additional supplementary figure (Fig. S10) showing the requested alignment and the conservation of key interacting residues.

Minor Comments:

1. Line 165: the correct call is Fig 1c

We have changed the referencing of figures in the main text accordingly.

2. Line 170 the correct call is Fig 1b-e

We have changed the referencing of figures in the main text accordingly.

3. Line 279: “was in the low nanomolar range (Table S3)” according to the table the affinity is in the micromolar range.

We apologize for this mistake - the reviewer is right that this is in the micromolar range, and we have modified the text accordingly.

4. Figure S3: The concentration of IgG here is 10µg/mL, which is different from Figure 2b. Please comment on this.

We thank the reviewer for spotting this inconsistency. In the initial screening for antibody neutralization, IgGs were used at 10µg/ml as the highest concentration (as shown also in Fig. 1i + j). In contrast, the goal of the assay shown in Figure 2b was to minimize the antibody concentration while still observing at least 90% reduction of infection. This allowed minimization of the requirement for glycosylated pORF2 decoy as a competition reagent and precluded unspecific effects during staining of infected cells.

Reviewer #4: Summary

This paper by Ssebyatika et al., describes novel glycan sensitive antibodies that protect against Hepatitis E virus infection. They used antigen-specific single-cell sorting, next generation sequencing, structural biology, and functional and biochemical antibody characterization in vitro and in vivo to understand the antibody response induced by acute HEV infection in humans. Their

results can have implications for the design of a cross-protective HEV vaccine and highlight the potential of “glycan-sensitive” human antibodies as novel tool to prevent and possibly treat HEV infection.

Overall, the paper is clear and well written although some clarifications could be made. The paper should be of interest to readers in the field of virology, antibody, and vaccine design.

We very much appreciate the positive feedback from the reviewer.

Major comments

For the non-Hepatitis E virus experts, it would be helpful if the authors explain how the infectious particles do not have glycans that are present at some other stages and the differences between the naked vs the quasi-enveloped virions. It may be helpful to have schematic showing the different compartments and the different irons as well as the known antigenic sites.

We understand that we have not sufficiently introduced this complex topic. In the revised manuscript, we have therefore extended the introduction a bit to better illustrate the differences between individual pORF2 forms and highlight the complex assembly pathway (lines 116-123).

Need to clarify the signal peptide -it seems that the construct 1 shown in Figure S1 does not have a signal peptide but how is this protein expressed in S2 cells?

The absence of a signal peptide precludes translocation into the secretory pathway, but the protein will still be expressed in the cytosol. As explained in the methods section construct 1 is therefore purified from the cell lysate instead of the supernatant. We have further clarified this in the figure legend describing Fig. S1.

In fig S1, other potential glycosylation sites are shown - are they present? The authors only mention one glycosylation site. Mass spec could be done to confirm.

We thank the reviewer for this suggestion. In the pORF2 P domain only a single N-linked glycosylation site is present (N₅₆₂), explaining why we focus exclusively on this glycosylation site in the presented study.

Does the recombinant HEV vaccine (Hecolin) bind to the glycan sensitive abs described here?

As outlined in response to reviewer #2 (see above) we have now appended an additional supplementary figure (Fig S10) showing an amino acid alignment of the P domains used in this study that indicates the conservation of key interacting residues. This alignment, which includes the P domain sequence of the HEV

vaccine strain (GenBank Acc. No. BAG09239.1), reveals the key interacting residues to be conserved between the used P domains of GT1-4 and Hecolin. Together with the very broad binding of the P domain of all four human pathogenic genotypes as well as the rat HEV, this alignment indicates that our glycan-sensitive antibodies also bind the HEV vaccine strain.

Can the authors resolve glycans or at least proximal NAG in the crystal structure of the glycosylated dimer? This is not mentioned in the text.

This is indeed a very interesting question that we will try to address in the future. In the current manuscript, we have not presented a crystal structure of the glycosylated pORF2 dimer or a glycosylated P domain. The presented crystal structures represent the non-glycosylated P domain only, where we cannot observe electron density for glycans.

Minor comments

Line 70 GT 1 or GT1 –

We have now harmonized the nomenclature and used GT 1 throughout the revised manuscript.

Would be nice to see B cell sorting plots as supplemental.

As outlined in response to reviewer #2 (see above) we have now prepared an additional supplementary figure (Fig S2) showing the flow cytometry plots.

Lines 133-134: reads weird; X-ray crystallographic analyses showed that targeted neutralization epitopes involved N562 within a highly conserved region at the pORF2 P domain dimer apex.

We thank the reviewer for pointing this out – we have rephrased this sentence (Lines 135-137).

Fig 2a. What do the VH labels mean?

The heavy chain germline genes of glycan-sensitive antibodies are indicated above the bar plot and this information has now been appended to the legend to Figure 2a.

Fig3. Label N3 on the figure - can be hard to see otherwise.

We thank the reviewer for this suggestion, N-linked glycosylation site N3 is now labeled in a revised figure 3, and this is explained in the figure legend.

Only one scFv and one Fab shown on the dimer which is confusing even though the authors write this is done for clarity.

We understand the reviewer's point and we have considered several other possibilities to solve this issue. However, a figure that shows both antibody fragments in complex with the P domain would be so wide that hardly any details of the P domain remain visible. Therefore, we prefer to stick with the current type of figure. We have clarified this further in the legend to figure 3.

Maybe each promoter could be colored in different shades of grey to see better the dimer interface.

We thank the reviewer for this suggestion. We have modified Fig. 4 accordingly.

Line 210-211: Rewrite: to "suggesting that this group of antibodies do not share a common genetic signature."

We thank the reviewer for this suggestion. We have rephrased the sentence accordingly.

Would be nice to show BLI and SPR curves in supplemental.

As outlined in the response to a comment by reviewer #2 (see above), we have now prepared an additional supplementary figure (Fig. S4) showing representative chromatograms that served for the determination of kinetic binding parameters.

Lines 232-237: paragraph could be rewritten: It is unclear if this is done with sera or with monoclonal antibodies. Is the dimer/nab complex removed prior neutralization?

We thank the reviewer for pointing out this issue. We have rephrased this paragraph in the revised manuscript and believe that it is clearer now.

The case for escape mutations should be put into context of variability of this virus (compared to

others)

We thank the reviewer for this suggestion. The discussion of escape mutations should consider the variability of the viruses involved. Hepatitis E virus (HEV) has a mutation rate of about $\times 10^{-3}$ base substitutions per site per year, but maintains a stable antigenic structure with only one serotype (Engle et al., J Clin Microbiol, 2002; PMID 12454155 / Takahashi et al., Hepatol Res, 2004; PMID 15288013). This stability reduces the likelihood of escape mutations, facilitating effective vaccine and neutralizing antibody development.

In contrast, Hepatitis C virus (HCV) has a similar mutation rate but can be classified into multiple genotypes and subtypes, increasing the risk of escape mutations and complicating therapeutic strategies. Thus, HEV's antigenic stability contrasts sharply with HCV's high variability, impacting the potential for escape mutations and challenges for vaccine design.

We appended a paragraph on this subject into the concluding remark section of the revised manuscript.

Details on interacting residues are not shown in figures 3 and 4. That could be added.

As outlined in the response to a comment by reviewer #2 (see above) we have now appended an additional supplementary figure (Fig. S10) showing an amino acid alignment of the P domains used in this study and the conservation of key interacting residues.

Luebeck, Dezember 18, 2024

Rebuttal letter

We thank all reviewers for their constructive comments and their positive feedback, for which we provide a point-by-point response below:

Reviewer #1 (Remarks to the Author):

The revised manuscript has been improved. However, several issues remain.

1. Lines 162 and 475: authors state these numbers (456-660) are correct. However, according to the information in Fig. S1, the P domain used for B cell was C terminally truncated (456-606). Please double check.

We apologize for having omitted Fig. S1 in our last revision. We have now corrected this to reflect the accurate description in the text.

2. Lines 221-222: “highlighting that such a ‘glycan-sensitive’ antibody response can be induced in most patients”. This may be overstated. Only two patients were analyzed in this study and there was only one such mAb was identified in patient #61.

We thank the reviewer for the comment. We have rephrased the conclusion into “... and were induced in both tested individuals.” (line 219 in the revised manuscript).

3. Fig. S5: The neutralization is not efficient for most mAbs including p60.12, p60.15 and p61.15. It would be hard to claim that these mAbs are superior to existing mAbs.

We thank the reviewer for this important remark. Figure S5 shows the neutralization of pUC83-2-27, another gt3 HEV strain that is not adapted to the cell culture system, by our identified antibodies. We were also surprised by the observed reduced efficiency of our antibodies against this strain and can currently not explain this phenomenon. In our opinion, the fact that most of the tested nAbs neutralized the pUC83 virus less efficiently than the p6 virus suggests rather an assay-specific limitation than an epitope-specific reduced neutralization potency of specific nAbs. We have added a sentence into the revised manuscript to illustrate this result (lines 253-255).

We agree that a superiority to existing antibodies cannot be claimed without further

experiments and a direct comparison, and we intentionally did not do this in the presented manuscript. We made the general statement that the "... observed interaction of glycan-insensitive antibodies with the dimeric pORF2 decoy suggests a better performance of glycan-sensitive antibodies ..." (lines 459-461) and complement this statement with the remark that "further studies will be required to directly compare the protective efficacy of different antibodies *in vivo*" (lines 461-462).

4. Fig. 5a, lines 380-382: "...the kinetics of the increase in viral RNA correlated with the decrease in antibody titer, suggesting that the presence of sufficient bnAb quantities in the serum can control HEV infection and minimize virus shedding". This explanation is not satisfactory. All 3 animals in the antibody treated group received the same quantities of antibody and virus. The decline of antibody in the serum is not sufficient to explain why 1 out of 3 animals got infected especially when only the data from one animal is shown.

We understand that we have not sufficiently clarified this point. We do not claim that passive immunization with bnAb 60.1 induces a sterile immunity. In a passive immunization experiment as the one presented the outcome of the challenge is highly influenced by the amount of challenge virus injected and we thus believe that the infection of a small number of hepatocytes cannot be fully excluded despite the presence of nAbs. Such breakthrough events have previously been described in passive immunization experiments in human liver-chimeric mice using monoclonal antibodies targeting the hepatitis B virus S Ag (PMID 36748051) and the hepatitis C virus (PMID 18452146). The breakthrough event shown in Fig. 5A illustrates that the initial infection was successful, but – in contrast to the control animals - the spread of the virus was efficiently controlled as long as p60.1 was detectable in the serum. Spread of the virus as judged from excreted viral RNA in stool was detected only after the decline of human antibodies in serum. We have clarified this further in the revised manuscript (lines 379-385).

We thank the reviewer for the remark on antibody concentrations of the other treated animals and have appended those data into the revised figure.

5. Lines 417-419: It has been shown that quasi-enveloped particles are completely cloaked in host membranes. Epitope specificity and kinetic binding do not seem to provide a satisfactory explanation for the efficient neutralization of quasi-enveloped particles.

We thank the reviewer for this comment and agree that we cannot exhaustively explain the observed efficient neutralization of purified quasi-enveloped particles. We feel that parallels in biology and life cycle to the hepatitis A virus (HAV) are obvious and therefore hypothesized a similar mechanism of neutralization as

proposed for HAV. We have now added a paragraph to the concluding remarks further discussing this hypothesis and carefully rephrasing our conclusion drawn in the manuscript. We also clearly state that additional studies are required to investigate this hypothesis (lines 432 - 443).

6. Lines 421-422: "The efficient block of liver infection in mice pretrated with p60.1 suggests efficient neutralization of quasi-enveloped particles". Please clarify if the inoculum used for the animal experiments was naked or quasi-enveloped. If naked virus was used, the conclusion would not be convincing.

We are grateful for the comment of the reviewer and apologize for the confusion. As stated above, we have rephrased the paragraph on neutralization of quasi-enveloped particles (see reply to comment #5). We have also added the information to the materials and methods section that the challenge virus used for Fig. 5A was derived from cell lysate, i.e., was mostly naked HEV gt3 virus (lines 708-709).

7. Lines 453-454: Does it mean that glycan sensitive Ab recognize eHEV in the blood?

We appreciate the reviewer's insightful question. In that paragraph of the manuscript, we describe the detection of antigens in ELISA. Given that we utilize detergent in both the washing and blocking buffers, it is anticipated that the quasi-envelopes of the virions get solubilized and thereby expose the viral capsid for antibody binding. We clarified this further in the revised manuscript (lines 470-475).

8. Since the whole idea of this paper is that the glycan-sensitive antibody will perform better in patients than glycan-insensitive antibody, I still think that this idea needs to be vigorously tested *in vivo* where these two types of antibodies are compared in the presence and absence of the decoy.

Currently, it remains elusive whether the newly identified "glycan-sensitive" antibodies perform better *in vivo* than existing or newly-identified "glycan-insensitive" antibodies. We agree that this requires additional extensive *in vivo* testing and we therefore clarified the interpretation of the observed results (lines 459-462)

Reviewer #2 (Remarks to the Author):

The quality for publication has already been met. However, it is still recommended that the author approaches the HEV-specific binding capacity analysis of the 3,364 antibodies. If the specificity is too low, subsequent analyses of the sequence characteristics of so-called specific antibodies could mislead researchers working on HEV-related studies.

We appreciate the reviewer's constructive feedback. It is important to note that our focus in this study has primarily been on identifying potent neutralizing antibodies for potential use as prophylactic or therapeutic agents. Furthermore, the dataset we present is the first known collection of antibodies from individuals acutely infected with genotype 3 HEV. We acknowledge that we do not claim to have provided a comprehensive analysis of the antibody responses. The significance of assessing HEV-specific binding capacity is recognized, and we will take this into consideration in future research endeavors.

Reviewer #3 (Remarks to the Author):

The authors appropriately addressed most of the comments from the reviewers. The paper is a valuable addition to the scientific literature.

I have still few minor comments:

1) In new Figure S2 the gate is placed on double antigen negative population. Took me a while to understand that those were the cells not sorted. Would be better to include another gate/shape to highlight what was actually sorted.

We thank the reviewer for this suggestion. We have now added such an additional shape to highlight the sorted cells and have explained this in the figure legend.

2) I am still of the opinion that the animal experiments should have been repeated, as these two experiments show different protection mechanisms, but the editor can make the call according to the journal policy. If experiments are not repeated, I think the limitation should be more clearly discussed within the paper.

We understand the reviewers concern and we have indeed repeated the co-housing experiment with six additional mice (two donor mice, two p60.1-treated mice and two control mice). The results of this experiment revealed the same protective capacity of the p60.1 bnAb as observed before, i.e., the p60.1 bnAb fully

protected the two mice from HEV infection in the co-housing setting, whereas both control mice were readily infected. These confirmatory results have now been added to the revised Fig. 5B. In addition, we have appended a paragraph to discuss the limitations of the available *in vivo* data (lines 405-420). In particular, we agree with the reviewer that at least from the presented data we cannot conclude whether both *in vivo* experiments highlight the same neutralization mechanism or two different neutralization mechanisms. This is partly due to the fact that the pathway by which the transmitted naked particles arrive in the newly infected liver remains elusive to date.

3) New Table S1. The resolution is too poor to read

We apologize for the poor quality of the table and have implemented a higher quality version into a revised supplementary material.

Reviewer #4 (Remarks to the Author):

The authors have addressed the comments.

We thank the reviewer for the positive feedback